# REPRESENTING LOCAL PROTEIN ENVIRONMENTS WITH MACHINE LEARNING FORCE FIELDS

**Meital Bojan**[*,1]**, Sanketh Vedula**[*,4,5]**, Advaith Maddipatla**[1,2,3]**, Nadav Sellam**[1]**,
Anar Rzayev**[1]**, Federico Napoli**[1]**, Paul Schanda**[1]**, Alex Bronstein**[1,2]

[1]IST Austria  [2]Technion, Israel  [3]University of Oxford, UK
[4]Princeton University  [5]Broad Institute of MIT and Harvard
[*]Equal contribution

## ABSTRACT

The local structure of a protein strongly impacts its function and interactions with other molecules. Representing local biomolecular environments remains a key challenge while applying machine learning approaches over protein structures. The structural and chemical variability of these environments makes them challenging to model, and performing representation learning on these objects remains largely under-explored. In this work, we propose representations for local protein environments that leverage intermediate features from machine learning force fields (MLFFs). We extensively benchmark state-of-the-art MLFFs, comparing their performance across latent spaces and downstream tasks, and show that their embeddings capture local structural (e.g., secondary motifs) and chemical features (e.g., amino acid identity and protonation state), organizing protein environments into a structured manifold. We show that these representations enable zero-shot generalization and transfer across diverse downstream tasks. As a case study, we build a physics-informed, uncertainty-aware chemical shift predictor that achieves state-of-the-art accuracy in biomolecular NMR spectroscopy. Our results establish MLFFs as general-purpose, reusable representation learners for protein modeling, opening new directions in representation learning for structured physical systems. Code and data are available at `https://github.com/mb012/MLFF_representation`.

## 1 INTRODUCTION

Proteins[1] are complex three-dimensional structures composed of hundreds to thousands of atoms, arranged in a specific manner in space. Local environments within a protein are highly diverse due to variability in the amino-acid sequence and the folding of the chain into a 3D structure. These factors govern a protein's functional mechanisms such as ligand binding, catalysis, and allostery, and make proteins uniquely challenging compared to small molecules. Learning compact and transferable representations of local protein environments is a central challenge for performing machine learning (ML) over biomolecules: the difficulty lies in jointly encoding the local chemical context, including atomic identities, bonds, and subtle biochemical properties, into a consistent and generalizable representation that can transfer across diverse protein modeling tasks.

Classical approaches employ hand-crafted descriptors that partially capture this information by embedding features such as dihedral angles, hydrogen bonds, or electrostatic terms, which often limits generalization across proteins and tasks. In computational chemistry, descriptors like Parrinello-Behler symmetry functions (Behler, 2011; Behler & Parrinello, 2007; Behler, 2016) are widely used to represent molecular environments (Jäger et al., 2018). These methods encode atomic interactions and geometry into concise, invariant representations suited for geometric modeling. Modern atomistic ML techniques (Schütt et al., 2018; 2021) implicitly learn similar representations with neural networks, achieving accuracy comparable to density functional theory (DFT) (Deng et al., 2023) and are used for molecular dynamics (MD) simulations. Recent advances have shifted neural-network interatomic

---

[1]Proteins are polymers of amino acids (also called residues). Residues share a common backbone of four heavy atoms (N, CA, C, O) and differ in their side chains.

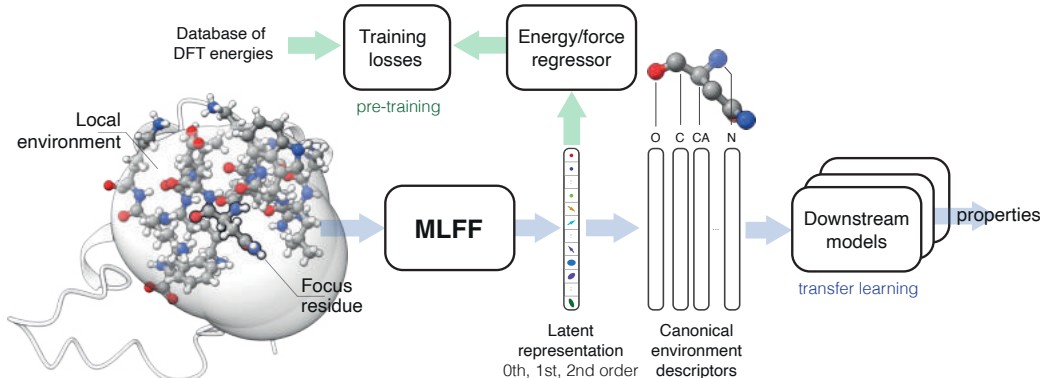

Figure 1: **Proposed construction of canonical local protein environment descriptors and their use.** Machine learning force field (MLFF) models are pre-trained as energy and force regressors on databases of DFT-calculated energies, enabling them to learn zeroth, first, and second order latent representations of interatomic interactions. We embed local protein environments by extracting latent embeddings from pre-trained MLFFs for a focus residue and all atoms within a 5Å radius of the residue. These embeddings are then mapped onto the atoms of the focus residue to construct canonical environment descriptors which can be used in downstream models for transfer learning to predict diverse chemical properties.

potentials from element-specific models to universal machine learned force-fields (MLFFs) trained on large datasets of molecules with DFT-simulated energies. Modern MLFFs train large neural networks to achieve DFT-level accuracy on millions of calculations, with representative families including AIMNet (Smith et al., 2018), MACE (Batatia et al., 2022; Kovács et al., 2025), and OrbNet (Qiao et al., 2020).

**Contributions and Summary of Results.** The key idea of this work is to repurpose MLFF embeddings from their conventional role in energy and force prediction to representing local chemical environments in proteins. We show that MLFFs learn compact and generalizable representations of atomistic systems that can be reused for a variety of downstream tasks. MLFF embeddings, unlike sequence-based representations (e.g., ESM (Lin et al., 2023)), are trained on quantum-mechanical data and encode physically grounded features such as bond geometry, torsions, and electronic interactions. Because these embeddings are defined per atom, they naturally transfer to unseen proteins. The atomic building blocks remain the same even when sequences or folds differ, enabling generalization to rare chemistries and out-of-distribution conformations. Analogous to foundation models in vision and language, these representations enable zero-shot and transfer learning, support data-driven priors over biomolecular environments, and position MLFFs as foundation models for structural biology. Outlined below are our major contributions and a summary of results. To assist the reader, we provide a succinct summary of the research questions, experimental designs, and key findings in Table B.2.

*Canonical local environments and MLFF representations.* Protein environments vary widely, making their direct comparison challenging. To make MLFF representations comparable across residues and proteins, we introduce *canonical environments*: regions centered on a *focus residue* and containing every amino acid whose atom coordinates lie within 5Å (Hausdorff distance) of a focus residue. We then construct transferable *environment representations* from the atom-wise representations of constituent atoms (or a subset thereof). To our knowledge, this is the first use of MLFFs to characterize local protein structure.

*MLFF representations effectively capture local protein structure and chemistry.* By analyzing over 165K protein environments extracted from 1048 non-redundant chains, we demonstrate that MLFF embeddings meaningfully organize biochemical information without any additional training: unsupervised clustering of these embeddings reveal secondary structure features and amino acid identities (see Fig. 2). Furthermore, we use these embeddings for transfer learning by training networks to predict protonation states calculated with a Poisson-Boltzmann solver (Reis et al., 2020). The resulting models based on MLFF embeddings – especially our best AIMNet-feature model – achieve the lowest errors on simulated $pK_a$ values, outperforming classical frameworks such as pKa-ANI (Gokcan & Isayev, 2022) and PropKa (Olsson et al., 2011), as well as GCNs built on learned or ESM-derived embeddings (see Table 1).

***MLFF representations allow computation of similarity metrics and calculating likelihoods.*** We show that MLFF embedding space is well-structured. Specifically, distances between environment embeddings reflect both geometric similarity and chemical context, enabling intuitive clustering of related local motifs. We construct likelihood and conditional likelihoods in the MLFF embedding space and show that they effectively capture the distribution of biomolecular environments (see Fig. A.7, A.11). We show that the defined likelihoods capture subtle structural deformations in protein structure and allow capturing distribution shifts, making them valuable for structural quality assessment, uncertainty estimation, anomaly detection, dataset curation, and potentially in guiding Boltzmann generators (Noé et al., 2019). Furthermore, leveraging the MLFF embedding space, we introduce a similarity metric to compare protein environments (see Appendix H.2).

***MLFFs enable transfer learning of physics-grounded, uncertainty-aware chemical shift predictor.*** In the context of biomolecular nuclear magnetic resonance (NMR) spectroscopy, we leverage MLFF embeddings and train downstream networks to predict protein chemical shifts, and demonstrate that this approach outperforms the current state-of-the-art predictor UCBShift2-X for both backbone and side-chain nuclei (see Fig. 3). We conduct a case study examining the ring current effects produced by aromatic side chains on chemical shifts of neighboring nuclei, and demonstrate that our predictor follows physically consistent trends, whereas UCBShift2-X shows unphysical behavior (see Fig. 4). We also pair the predictor with MLFF-derived likelihoods to assign confidence scores, yielding a reliable uncertainty measure around the predicted shifts (see Fig. A.7).

***Interpreting MLFF representations.*** We probe the content of MLFF embeddings in two controlled settings. Firstly, we systematically modify side chain orientations and track how the embeddings respond (see Fig. 4 and 6). Secondly, we follow an unfolding simulation in which a seven-residue $\alpha$-helix extends into a strand (see Fig. A.9, A.8, and A.10). In both cases, principal-component analysis uncovers clear latent directions that mirror the underlying structural changes. Finally, to understand the information contained within the MLFF embeddings, we ask the following question: *can we invert the MLFF embedding to recover the underlying local protein environment?* Casting this question as an inverse problem, we *guide* (Maddipatla et al., 2025; 2024; Levy et al., 2025) AlphaFold3 (Abramson et al., 2024) and reasonably recover protein conformations underlying a target MLFF embedding. This partial recovery suggests that MLFF embeddings encode necessary information required to invert a local protein conformation (see Appendix. J.2).

***Benchmarking different MLFFs.*** We benchmark three model families – MACE, OrbNet, and AIMNet – on tasks defined on protein environment that include secondary structure assignment, amino acid identification, acid dissociation constant ($pK_a$) and chemical shift prediction (see Tables B.4, B.3, B.5, B.6). MACE-based embeddings perform best on every task except $pK_a$ prediction, where AIMNet exhibits superior performance (see Table 1).

Together, these steps recast MLFFs as general-purpose biochemical feature extractors for proteins, providing canonical representations, principled similarity/likelihood tools, and state-of-the-art downstream performance with calibrated uncertainty. The remainder of the paper is organized as follows. Section 2 introduces MLFFs and reviews closely related work, and Section 3 describes the construction of canonical local environments. Sections 4-6 demonstrate that the proposed representations capture protein structure and chemistry and enable state-of-the-art predictors. Section 7 interprets the learned representations, and Section 8 outlines limitations and future directions. Due to space constraints, the Appendix contains a significant amount of supplementary material, including additional figures, tables, experiments, an extended background, and full experimental details. We encourage the reader to consult the table of contents in the Appendix for a complete overview.

## 2 BACKGROUND & RELATED WORK

**MLFFs.** Machine learning force-fields (MLFFs) are neural networks trained to reproduce quantum-mechanical potential energy surfaces. Given atomic coordinates and elements as inputs, MLFFs apply symmetry-preserving encodings (e.g., local environments, equivariant basis functions), and output per-atom energy contributions whose sum yields the total potential energy. Consequently, forces are obtained as the negative energy gradient, enabling molecular dynamics (MD) simulations with near-quantum accuracy at orders of magnitude lower cost than direct DFT calculations. In this work, we focus on three representative MLFF families with distinct design principles: **MACE** (Batatia et al., 2022) and **Egret** (Wagen et al., 2025) combine equivariant message passing with high-order

Table 1: **Acid Dissociation constant (pK$_a$) prediction accuracy.** Reported are mean absolute errors (MAE) with respect to pK$_a$ values estimated using the Poisson–Boltzmann solver PypKa (Reis et al., 2020) on protonating / deprotonating amino acids. The table compares (i) classical pK$_a$ predictors (Olsson et al., 2011; Gokcan & Isayev, 2022), (ii) a GCN with learned embeddings ("Learned"), (iii) ESM-based features + GCN (Ouyang-Zhang et al., 2025; Hayes et al., 2025; Lin et al., 2023), and (iv) MLFF-based features + GCN (ours). PropKa and pKa-ANI were originally trained on experimental pK$_a$ values and are therefore not specifically optimized for the computational reference used here.

The best performing model for each metric is indicated in **bold**.

| Residues | # Samples | PropKa | pKa-ANI | Learned | ESM-Features + GCN | | | | MLFF-Features + GCN (Ours) | | | |
| | | | | | ESM3 (Seq) | ISM | ESM (Struc) | ESMFold | MACE | OrbNet | AIMNet | Egret |
| --- | --- | --- | --- | --- | --- | --- | --- | --- | --- | --- | --- | --- |
| Glutamic acid | 12592 | 0.551 | 0.445 | 0.518 | 0.459 | 0.382 | 0.360 | 0.351 | 0.306 | 0.306 | **0.265** | 0.304 |
| Aspartic acid | 9841 | 0.469 | 0.473 | 0.582 | 0.528 | 0.430 | 0.388 | 0.419 | 0.280 | 0.284 | **0.267** | 0.272 |
| Lysine | 12009 | 0.393 | 0.401 | 0.652 | 0.359 | 0.304 | 0.295 | 0.278 | 0.320 | 0.282 | **0.270** | 0.298 |
| Histidine | 1182 | 0.561 | 0.426 | 0.615 | 0.488 | 0.428 | 0.408 | 0.383 | 0.424 | 0.441 | **0.380** | 0.440 |

many-body interatomic potentials; **OrbNet** (Qiao et al., 2020) augments graph neural networks with semi-empirical orbital features to achieve DFT-level accuracy with reduced data requirements; and **AIMNet** (Zubatyuk et al., 2019) extends ANI with learned atomic embeddings and multitask training (energies, charges, spin), improving transferability to charged and open-shell systems. While these models achieve remarkable accuracy for small molecules, materials, and bipeptides, MLFFs continue to face challenges in capturing long-range interactions in larger biomolecular systems like proteins. Further details on these models and a short background on MD and DFT is provided in Appendix D.

**Related work.** Recent work has highlighted the versatility of pretrained MLFFs as learned representations for molecular tasks. Shiota et al. (2024) use MACE and M3GNet features to predict chemical shifts in small molecules, while Elijošius et al. (2025) employ MACE embeddings for zero-shot molecular generation via evolutionary search. However these works are limited to small-molecules and do not scale to large, structurally heterogeneous biomolecules like proteins. Similar to our idea, Gokcan & Isayev (2022) extract ionizable residues and use ANI-2x to construct per-residue atomic environment vectors (AEVs), which are then used as input to train a regressor for pK$_a$ prediction. However, these AEVs are fixed, hand-crafted descriptors derived from symmetry functions that do not use message passing over the local environment. As a result, they lack the ability to adaptively capture the context-dependent interactions present in protein systems. In contrast, our descriptors encode a richer biochemical context. Moreover, we train a GCN over this many-body embedding space, enabling a context-aware model. As shown in Table 1, our method achieves lower pK$_a$ prediction error and generalizes to additional biochemical tasks beyond pK$_a$.

## 3 Representing protein local environments with MLFFs

**Notation.** We denote a residue in a protein as $a$, and the all-atom representation of the protein structure as $\mathcal{X} = \{(\mathbf{x}_1, z_1) \ldots (\mathbf{x}_m, z_m)\}$, where $\mathbf{x}_j \in \mathbb{R}^3$ and $z_j$ are the location and atomic number of $j^{th}$ atom in $\mathcal{X}$, respectively. Let $f_\theta : \mathcal{X} \to \mathcal{Y}$ be the MLFF with parameters $\theta$, and $\mathcal{Y} = \{(\mathbf{y}_1), \ldots, (\mathbf{y}_m)\}$ the atom-wise embeddings of the MLFF, where $\mathbf{y} \in \mathbb{R}^d$. The *local environment* of a *focus residue* $a$ is denoted by $\mathcal{X}_a \subseteq \mathcal{X}$. The subset of atoms in residue $a$ used for predicting biochemical properties is denoted as $\mathcal{A}_a$. Given a subset of atom indices $\mathcal{A}_a$, we denote by $\mathcal{Y}_{\mathcal{A}_a} = \{(\mathbf{y}_i) \mid i \in \mathcal{A}_a\} \subseteq \mathcal{Y}$ the restriction of $\mathcal{Y}$ to those atoms.

We seek a representation of a local protein environment to be (i) sensitive to local changes, (ii) insensitive to global variations, (iii) fast to compute, (iv) canonical, enabling direct comparison across diverse environments, and (v) effective and generalizable to unseen environments. Encoding full proteins with several thousand atoms with MLFFs is computationally inefficient and redundant since we are solely interested in locally sensitive features. To construct such representations, we propose to construct a local environment around a focus residue to be encoded later by the MLFF.

**Constructing local environments.** To balance computational efficiency while retaining local structural context, given a focus residue $a$, we construct the environment $\mathcal{X}_a$ as the union of all residues whose atoms are at most 5Å away from the atoms of $a$. The procedure is described in App. Alg. 1, and an ablation over the choice of radius is provided in Appendix F.

**MLFF representations.** Given an environment $\mathcal{X}_a$, MLFFs produce atom-level features over the layers of the network. These learned atomistic features are *contextual*, influenced not only by the

atom alone but also by the other atoms in the environment due to the message-passing operations within MLFFs. Given $\mathcal{X}_a$, atom-wise feature representations are extracted from the final layer of the MLFF.[2] Each MLFF produces representations of shape $N \times d$, where $N = |\mathcal{X}_a|$ and $d$ is the dimension of the representation. To obtain *canonical* representations that are comparable across residues, we retain only $\mathcal{Y}_{\mathcal{A}_a}$, the representations corresponding to atoms in $\mathcal{A}$. Given a protein structure, this process is applied to each residue in the sequence to obtain residue-wise features. A visual depiction of this procedure is presented in Fig. 1.

**Data.** To ensure a consistent dataset across all experiments, we sourced data from RefDB, a curated subset of the Biomolecular Magnetic Resonance Databank (BMRB). The final dataset includes 1048 BMRB entries, filtered for sequence redundancy and split into 823 training and 225 test proteins. From these structures, a total of 165K local protein environments were extracted using a 5Å radius threshold based on Hausdorff distance. We use AlphaFold2 (Jumper et al., 2021) predicted 3D structures for all sequences, followed by hydrogen addition and Amber99 force-field relaxation (Hornak et al., 2006). We refer to Sec. E for a detailed justification of our choice, and for further details on data curation.

**Baselines** To evaluate the utility of MLFF embeddings, we compare the performance of MLFF embeddings + GCN against different embedding baselines: (i) sequence-model embeddings (ESM3 (Hayes et al., 2025)) and sequence models with enhanced structural representations (Ouyang-Zhang et al., 2025) + GCN, (ii) ESMFold (Lin et al., 2023) structure-model embeddings + GCN, (iii) hand-crafted statistics-based descriptors (LOCO-HD) (Fazekas et al., 2024) + GCN, and (iv) simple learned embeddings + GCN. In the learned-embedding baseline, the GCN is trained end-to-end from atom types and 3D Cartesian coordinates from AlphaFold2 Jumper et al. (2021). The coordinates are first mapped to a sinusoidal positional encoding, which is then concatenated with the embedded atom-type identifiers to produce the node features for the GCN.

## 4    MLFF REPRESENTATIONS CAPTURE LOCAL PROTEIN STRUCTURE AND CHEMISTRY

In what follows, we first present a zero-shot analysis of MLFF representations to assess whether they inherently capture biochemically meaningful information without task-specific training. We then apply transfer learning by training neural networks on top of the frozen embeddings to predict structural and chemical properties. We first consider amino acid identity and secondary structure prediction as motivating examples, and then progress to more challenging downstream regression tasks such as pK$_a$ and NMR chemical shift prediction.

**Amino acid identity & Secondary structure.** A protein's *primary structure* is its linear sequence of amino acids linked by covalent peptide bonds and implicitly encode the 3D protein structure. Its *secondary structure* is a local motif of backbone conformation, stabilized by hydrogen bonds between non-adjacent residues (Kabsch & Sander, 1983), with the most common motifs being $\alpha$-helices (H) and $\beta$-sheets (E).

To examine the information encoded by the MLFF representations in a zero-shot setting, the atomistic features from MACE were projected into a two dimensional space using Uniform Manifold Approximation and Projection (UMAP) (McInnes et al., 2018). The resulting embeddings, shown in Fig. 2, are annotated with amino acid labels, secondary structure, and dihedral angles. Notably, features corresponding to $\alpha$-helices and $\beta$-sheets form distinct clusters in the UMAP space. A similar pattern of separation is observed when embeddings are annotated by amino acid identity and backbone dihedral angles, indicating that chemically and structurally distinct features are well-represented in the feature space.

To quantitatively assess this capability, we perform lightweight transfer learning by training classifiers to predict the secondary structures and the amino acid identities from the frozen embeddings. Further details on the loss function, target label space, and the definition of $\mathcal{A}_a$ can be found in Appendix G.1, while a comprehensive description of the model architecture is provided in Appendix G.3. Classifier accuracy was evaluated using precision, recall, and F1 scores. Results for secondary structure prediction are summarized in Table B.3, while amino acid classification results are presented in Table B.4. For secondary structure, models trained with MACE and Egret features consistently

---

[2]See Appendix F.2 for an ablation study comparing representations extracted from different MLFF layers.

achieve superior prediction on average. For amino acid prediction, models trained using Egret features consistently outperform those trained with alternative representations.

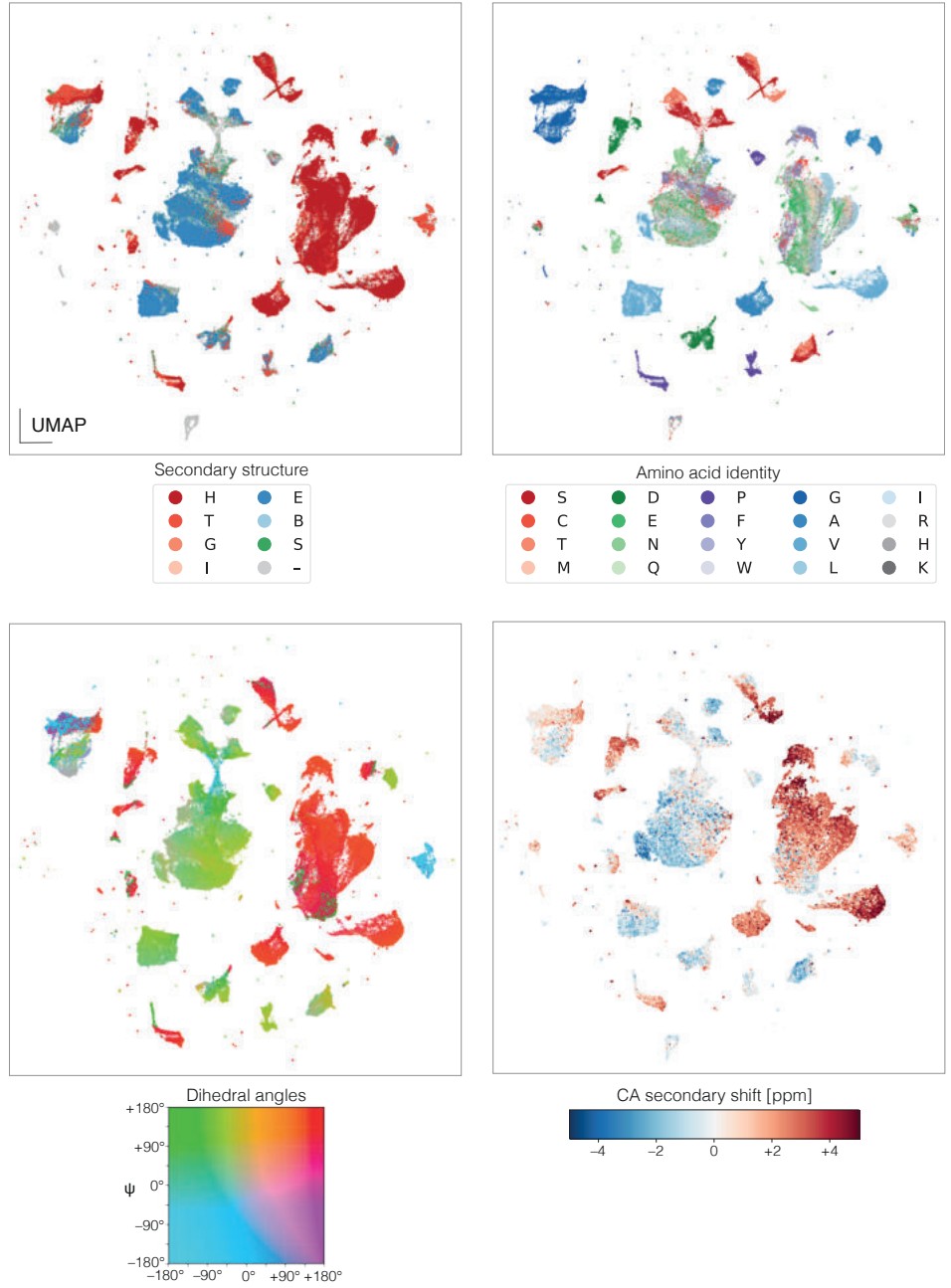

Figure 2: **MACE embedding space reveals meaningful structural and chemical information.** Depicted are two-dimensional UMAP coordinates of $165,913$ protein environments from $1327$ non-redundant chains predicted by AlphaFold2 (Jumper et al., 2021), labeled left-to-right, top-to-bottom according to the DSSP secondary structure class (Table B.12), amino acid chemical identity, the pair of backbone dihedral angles $(\phi, \psi)$, and CA secondary chemical shift (relative to a random coil).

**Acid dissociation constants ($pK_a$).** $pK_a$ quantifies the proton-donation propensity of a titratable group, i.e., its acidity. It is the pH at which the group is half protonated across the ensemble. In proteins, $pK_a$ values often deviate from their canonical solution values due to local electrostatics and solvent accessibility. These context-dependent shifts make $pK_a$ prediction a strong test of

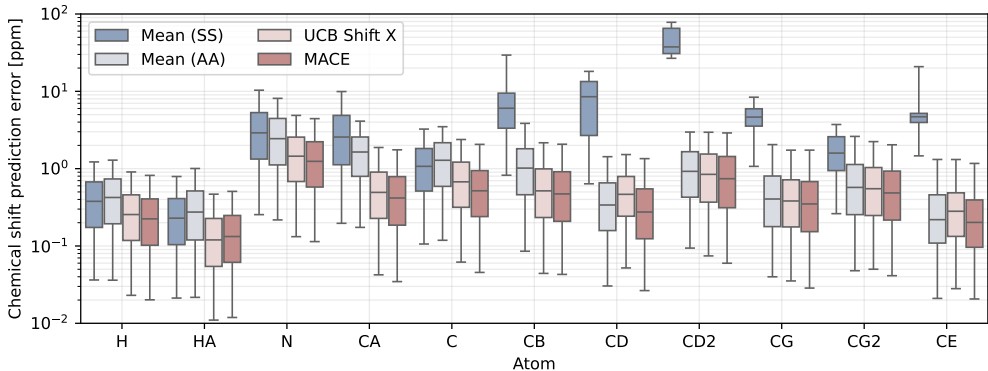

Figure 3: **Chemical shift prediction errors for different atom types** evaluated on a test set of $132,228$ environments from 203 non-redundant BMRB records with experimentally determined chemical shifts used as the reference. The median prediction error in ppm and the $25\% - 75\%$ (boxes) and $5\% - 95\%$ (whiskers) confidence intervals are depicted.

whether atomistic representations capture local chemical environments (see App. D for biochemical background).

For quantitative assessment, a regression model was trained on amino acids with ionizable, proton-transferring side chains to predict their pK$_a$ values. Ground-truth pK$_a$ values were estimated using a Poisson-Boltzmann solver (Reis et al., 2020). The evaluation focused on glutamic acid (GLU) and aspartic acid (ASP), which tend to deprotonate, and lysine (LYS) and histidine (HIS), which tend to protonate. As shown in Table 1, models trained with AIMNet features achieve the best overall performance among all feature types because it is trained to predict multiple molecular properties simultaneously. For additional comparisons with PropKA (Olsson et al., 2011) and pKa-ANI (Gokcan & Isayev, 2022) on experimental pK$_a$ values and PDB structures, as well as training details, see Appendix G.

## 5 LIKELIHOODS AND SIMILARITY OF LOCAL PROTEIN ENVIRONMENTS

The MLFF embedding space encodes local biomolecular structure, capturing structural and chemical context. We leverage the richness of MLFF embedding spaces to define a distribution over biomolecular environments. Given a set of reference environments $\mathcal{E}_{\text{ref}} = \{\mathcal{X}_1, \ldots, \mathcal{X}_n\}$ and a reference atom set $\mathcal{A}$, we define the likelihood of an environment $\mathcal{X}_a$ at a focus residue $a$ as

$$p(\mathcal{X}_a) = \frac{1}{|\mathcal{E}_{\text{ref}}|} \sum_{\mathcal{X}_{a'} \in \mathcal{E}_{\text{ref}}} \exp\left( -\frac{\|f_\theta(\mathcal{X}_a)|_{\mathcal{Y}_\mathcal{A}} - f_\theta(\mathcal{X}_{a'})|_{\mathcal{Y}_\mathcal{A}}\|^2}{2\sigma^2} \right), \quad (1)$$

where $\sigma$ refers to the *bandwidth*, and it controls the influence of each reference environment on $\mathcal{X}_a$. This is equivalent to performing kernel-density estimation in the MLFF embedding space, with a radial basis function kernel. Intuitively, the likelihood measures how typical $\mathcal{X}_a$ is among the reference environments $\mathcal{E}_{\text{ref}}$. Subsequently, the conditional likelihoods can be defined similarly to Eq. 1 by curating an appropriate set of reference environments that satisfy the chosen conditioning.

**Capturing distribution shifts.** To evaluate the quality of unconditional likelihood estimation and its ability to detect subtle distribution shifts in the local structure, we curated a set of reference environments randomly sampling $10,000$ environments from $1,100$ protein structures that were relaxed using the Amber99 force field (Hornak et al., 2006). We then measured the likelihoods of each environment sampled from a test set of 225 proteins, before and after performing Amber99 relaxation. The results shown in Fig. A.7 demonstrate that the likelihood function is sensitive to subtle conformational changes. Relaxed structures consistently receive higher likelihoods, and the distribution of the paired differences captures fine-grained structural variations. This makes the approach well-suited for detecting out-of-distribution conformations and assessing local structural quality.

**Secondary structure conditioned likelihoods.** To further evaluate the sensitivity of conditional likelihoods, we randomly sampled 1000 environments from secondary structures annotated as $\alpha$-

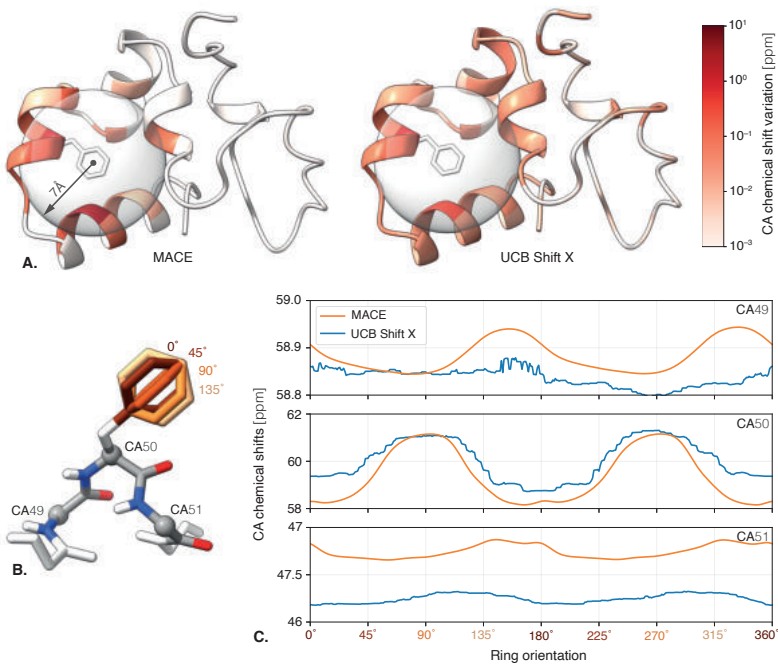

Figure 4: **Synthetic example showing the influence of a phenylalanine sidechain aromatic ring on surrounding chemical shifts. A.** The magnitude of change of backbone CA chemical shifts over different ring orientations as predicted using the proposed MACE-based predictor (left) and UCBShift-X (right). A 7Å sphere indicates the radius from the ring center at which the influence of the ring current is expected to become negligible. Note that UCBShift-X predicts much longer-range, albeit small, ring influence extending beyond 20Å. **B.** Locations of three nearby CA atoms; and **C.** their predicted chemical shifts vs. the ring orientation. Note the smooth $180°$-periodic behavior of the MACE shift prediction and the decay of the effect scale with the distance from the ring.

helix (H), $\beta$-strand (E), and turn (T) by DSSP, and constructed conditional likelihoods $p(\mathcal{X}|H)$, $p(\mathcal{X}|E)$, and $p(\mathcal{X}|T)$, respectively. Fig. A.11 depicts the projections of conditional likelihood triplets $(\log p(\mathcal{X}_i|E), \log p(\mathcal{X}_i|H), \log p(\mathcal{X}_i|T))$ of $8,163$ environments from the test set. The results highlight a clear separation by true secondary structure of the environment. The one-dimensional marginal distributions further emphasize the distinct statistical profiles of each structural class.

We further direct the reader to Fig. A.13 and Appendix H.2 for a detailed analysis of the use of MLFF embeddings to evaluate chemical environment similarity.

# 6 PHYSICS-GROUNDED AND UNCERTAINTY-AWARE CHEMICAL SHIFT PREDICTION FOR PROTEINS

**Background and prior art.** We refer an uninitiated reader to Appendix D for a short background on chemical shifts and biomolecular NMR. Several computational methods have been developed to predict NMR chemical shifts from molecular structures (Shen & Bax, 2008; Neal et al., 2003; Han et al., 2011; Li & Brüschweiler, 2012; 2015; Kohlhoff et al., 2009; Meiler, 2003; Moon & Case, 2007), with recent approaches increasingly relying on machine learning. The current state-of-the-art, UCBShift (Zhou et al., 2023), combines sequence and structure alignment with a random forest model, using reference proteins and structural descriptors to guide predictions. Its latest iteration, UCBShift 2.0 (Ptaszek et al., 2024), extends this framework to include side-chain atoms alongside backbone predictions. Despite their effectiveness, these methods depend on reference-based similarity measures, which limits their generalizability.

**MLFF-based shift predictor.** Chemical shift prediction is formulated as a transfer learning task over frozen MLFF embeddings, where a graph neural network takes the pretrained atomistic representations as input and predicts the chemical shift of a target atom. Separate models are trained for the N, CA,

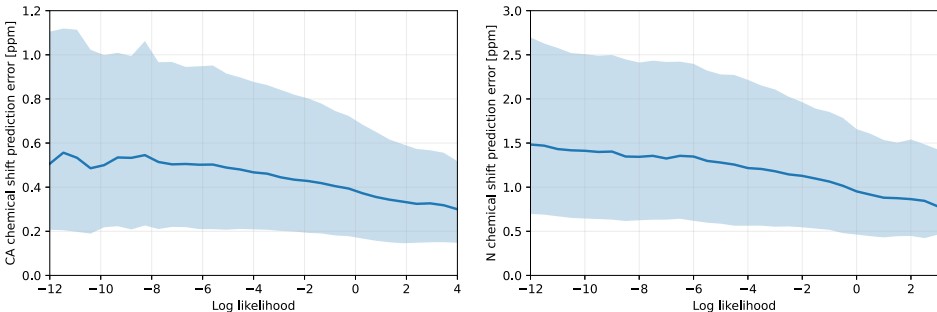

Figure 5: **Lower likelihood environments result in larger chemical shift prediction error.** Chemical shift prediction accuracy of CA (left) and N (right) atoms stratified by the KDE-estimated likelihood of the corresponding MACE descriptors. Depicted are the median and 25%-50% confidence intervals. Higher-likelihood environments correspond to lower prediction error and can be used as an uncertainty measure.

C, H, and HA backbone atoms, as well as the CB, CG, CD, CD2, CG2, and CE side-chain atoms. Dataset curation, model architecture, and training details are provided in Appendix E, G.3.

As evident from Fig. 3, the proposed predictor outperforms UCBShift2 for both backbone and side-chain heavy atoms, with the exception of the Hydrogen alpha atom. A comprehensive quantitative comparison of chemical shift prediction accuracy across different MLFF embeddings, ESM variants, and learned embeddings is provided in Tables B.5 and B.6. We observe that MACE-based models consistently achieve the best performance.

To further probe whether the shift predictor correctly captures local structural effects, we design three case studies: one is shown here, and the other two are provided in Appendix I for completeness.

**Effects of ring currents.** An external magnetic field induces a flow of delocalized $\pi$ electrons in an aromatic ring, which in turn produces its own magnetic field. Because chemical shifts depend on the net magnetic field produced by the local structural environment, they are strongly influenced by such ring currents. As the phenylalanine ring has a $C_2$ symmetry, states separated by rotation of the ring by $180°$ are indistinguishable. To probe whether our shift predictor captures ring-current effects accurately, we synthetically modified the side chain of a Phe (residue 50 in PDB ID: 1ZV6) by rotating its aromatic ring about the $\chi_2$ dihedral angle from $-180°$ to $180°$. Fig. 4 depicts the studied environments and the predicted chemical shifts. Our MLFF-based shift predictor shows the expected $180°$ periodicity (Haigh & Mallion, 1979) in the chemical shifts of backbone CA atoms in the vicinity of the ring, with the ring current influence decaying smoothly as the distance from the ring increases and becoming negligible after 7Å. In contrast, UCBShift extends this influence beyond the expected range and fails to reproduce the smoothness and periodicity expected in theory.

Next, we investigate whether the aforementioned likelihood computation for an environment can reliably anticipate the resulting chemical shift prediction error.

**Uncertainty estimation.** We first build the unconditional likelihood model $p(\mathcal{X})$ for biomolecular environments on the training set used for shift prediction. We then evaluate this likelihood for every test environment and record the corresponding chemical shift prediction error. Fig. 5 reports the distribution of errors, stratified by likelihood. Lower likelihood environments yield higher errors, making likelihood a practical *confidence* measure representing, to the best of our knowledge, the first such score reported for chemical shift prediction.

We summarize by concluding that MLFF-based shift prediction not only delivers better accuracy, but is also physics-grounded and allows us to provide confidence estimates for each prediction.

## 7 INTERPRETING MLFF REPRESENTATIONS

While MLFF atomistic features excel at representing local protein structure and chemistry, it remains unclear how they respond to structural perturbations or encode meaningful physical properties. To address this, the behavior of MLFF representations under specific conformational changes is

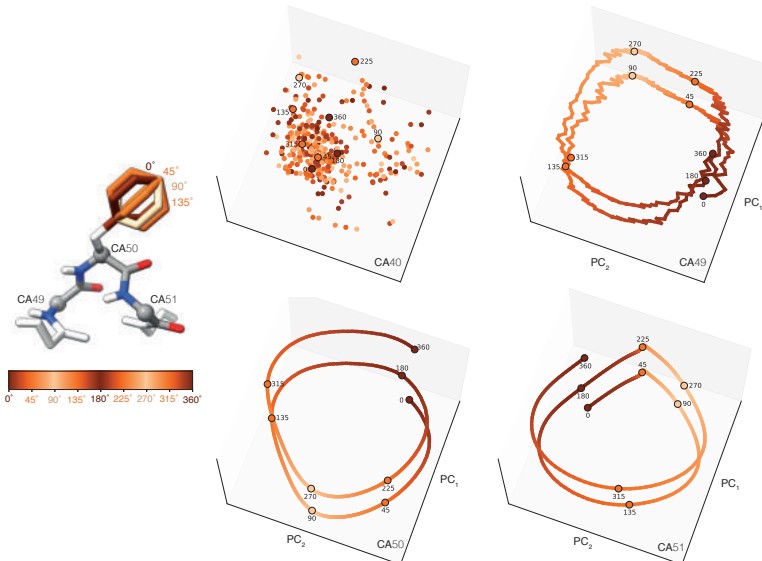

Figure 6: **Structure of the MACE embedding space for the rotating phenylalanine aromatic ring from Fig. 4**. Left: a fragment of a protein with simulated rotation of the phenylalanine sidechain ring. Right: Mace embeddings of backbone CA atoms projected onto the first two principal components. The vertical axis separates the different rotation angles for visual clarity. Note that residues close to the ring (50 and 51) exhibit a one-dimensional structure with nearly perfect 180-degree periodicity. The farther residue 49 shows some breakdown of this structure, while residue 40, uninfluenced by the ring current, manifests a lack of relation between ring rotation and the embedding space structure.

examined. Such changes induce smooth, interpretable trajectories in the embedding space, suggesting that the features capture physical motions and respect locality, continuity, and symmetry.

**Rotated phenylalanine sidechain aromatic ring.** Using the synthetically modified phenylalanine, we perform PCA on MACE embeddings obtained at every $\chi_2$ angle for the nearby CA atoms (residues $50$ and $51$ of PDB ID: 1ZV6). We observe that the first two principal components trace a smooth, one-dimensional curve, capturing the expected $180°$ periodicity of the rotation. However, the effect diminishes with distance: residue 49 begins to deviate from the periodic pattern, while distant residue 40 exhibits no discernible structure, illustrating the spatial locality of the descriptor's sensitivity.

Additional case studies further analyzing the embedding space for helix-to-strand unfolding MD trajectory and experiments related to inverting MLFF embeddings, are presented in Section I and Section J.2, respectively, in the Appendix.

## 8 DISCUSSION

In this work, we introduced a novel way to represent local protein environments by repurposing embeddings from MLFFs. To our knowledge, this is the first demonstration that MLFF latent spaces – trained exclusively on small-molecule quantum data – organize according to meaningful biochemical factors such as secondary structure, residue identity, protonation state, and chemical shift. We further show that these embeddings can be directly reused for diverse protein modeling tasks without retraining. This establishes MLFFs as general-purpose, reusable representation learners for structural biology and opens a new paradigm for leveraging pretrained molecular models in this domain.

While our downstream models used frozen MLFF embeddings, task-specific fine-tuning of MLFFs is a promising direction that may yield further gains. Moreover, our chemical shift prediction for HA, while accurate, still slightly lags behind UCBShift, pointing to opportunities for future improvement. Notwithstanding these limitations, we believe that the proposed fully differentiable chemical shift predictor can be used to guide AlphaFold and similar generative models (Maddipatla et al., 2025) for determining structure from experimentally measured chemical shifts – an important task in protein NMR that has so far been hindered by the complexity of structure-to-chemical shift relation.

## 9 ETHICS STATEMENT

All authors have read and complied with the ICLR Code of Ethics. This work does not involve human subjects, personally identifiable information, or sensitive data. All datasets mentioned in Appendix E are publicly available, and we adhered to best practices to ensure privacy, fairness, and transparency. We do not foresee any direct ethical risks associated with the methods or results presented.

## 10 REPRODUCIBILITY STATEMENT

The code, trained models, and data-processing scripts are publicly available at `https://github.com/mb012/MLFF_representation`. In addition, complete details of the models and optimization parameters are provided in Appendix G.3. The hardware resources used to produce the results are specified in Appendix I.4. The loss functions, evaluation metrics, and details regarding ablation studies are specified in Appendix G. These details ensure that all results reported in the paper can be independently verified.

### ACKNOWLEDGMENTS

This work was supported by the Institute of Science and Technology Austria (ISTA) through the IPC grant "Generative Protein NMR" and by the Israeli Science Foundation (ISF) under grant number 1834/24. This research used resources of the Institute of Science and Technology Austria's scientific computing cluster. S.V. was supported in part by funding from the Eric and Wendy Schmidt Center at the Broad Institute of MIT and Harvard.

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

APPENDIX

# A Summary of Experiments

Table B.2: **Experimental overview.** Summary of key research questions, experimental design, baselines, main takeaways, and supporting evidence. "ESM family" includes embeddings from ESM2/ESMFold (Lin et al., 2023), ISM (Ouyang-Zhang et al., 2025), and ESM3 (Hayes et al., 2025). On the other hand, "learned" refers to the learned-embedding baseline described in Section 3.

| Question | Experiment Design | Baselines | Takeaway | Evidence |
|---|---|---|---|---|
| **1. Do MLFF representations capture structural and chemical properties?** | Train classifiers to predict amino acid identity (20 types) and secondary structure ($\alpha$-helix, $\beta$-strand, other) from MLFF embeddings. Visualize embedding space using UMAP. | MACE, Egret, OrbNet, AIMNet, ESM family | Egret best for amino acid classification; MACE/Egret best for secondary structure. UMAP shows clear clustering by chemical identity and structural motifs. | Fig. 2 |
| **2. Can MLFF representations predict $pK_a$ values?** | Train regression models to predict $pK_a$ for ionizable residues (GLU, ASP, LYS, HIS) using MLFF embeddings. Target $pK_a$ values are approximated by PypKa's Poisson-Boltzmann solver. | PropKa, pKa-ANI, ESM family, learned, and MLFF variants | AIMNet features achieve best performance. | Table 1 |
| **3. Can MLFF embeddings detect distribution shifts and separate structural classes?** | (a) Measure likelihood of environments before/after Amber99 relaxation using kernel density estimation. (b) Construct conditional likelihoods for different secondary structures. | Compare distributions before/after relaxation | (a) Relaxed structures receive higher likelihoods. (b) Conditional likelihoods clearly separate secondary structure classes. | Figs. A.7, A.11 |
| **4. Can MLFF-based predictors outperform state-of-the-art chemical shift prediction?** | Train GNNs on MLFF embeddings to predict NMR chemical shifts for backbone (N, CA, C, H, HA) and side-chain atoms. | UCBShift2-X (SOTA), ESM family, learned | MLFF-based predictors outperform UCBShift2-X for backbone and side-chain heavy atoms (except HA). MACE-based models perform best overall. | Fig. 3, Table B.5, Table B.6 |
| **5. Are MLFF-based chemical shift predictions physically consistent?** | Three case studies: (1) Systematic phenylalanine ring rotation (0-$2\pi$) to probe ring-current effects. (2) MD simulation of helix-to-strand transition. (3) Alternative conformations in protein 4OLE:B. | UCBShift2-X, experimental chemical shift trends | MLFF predictor shows expected $\pi$-periodicity, correct C$\alpha$/C$\beta$ shifts during unfolding, and distinguishes conformational states. UCBShift2-X shows unphysical behavior. | Figs. 4, A.8, A.12 |

**Table B.2 (continued)**

| Question | Experiment Design | Baselines | Takeaway | Evidence |
|---|---|---|---|---|
| **6. Can likelihood serve as an uncertainty estimate?** | Build unconditional likelihood model on training embeddings; correlate environment likelihood with chemical shift prediction error on test set. | No direct baseline (novel metric) | Low-likelihood environments consistently exhibit higher prediction errors, supporting use for uncertainty estimation. | Fig. 5 |
| **7. How do MLFF embeddings respond to structural perturbations?** | Perform PCA on MACE embeddings during: (a) systematic phenylalanine ring rotation, (b) helix-to-strand MD simulation. Analyze embedding trajectories and spatial locality. | Visual analysis of embedding trajectories | Embeddings trace smooth trajectories reflecting structural changes, show expected periodicity, and reveal interpretable latent directions. | Figs. 6, A.9, A.10 |
| **8. Can MLFF embeddings be inverted to recover structure?** | Optimize AlphaFold3 structures using only MACE descriptor matching as loss function. Test on alternate conformations in: 4OLE,3V3S,4NPU,5G51, 7EC8. | Direct comparison to target conformations | Backbone geometry recovered with high accuracy; side-chain orientations less precise. MLFF embeddings encode most but not all information needed for reconstruction. | Fig. A.14, A.15, A.16 |
| **9. Comparative benchmarking and ablation studies** | (1) Benchmark all MLFF families on all tasks. (2) Ablate atom set selection. (3) Compare MLFF similarity metric to LoCoHD. | MACE, Egret, OrbNet, AIMNet variants; LoCoHD metric | MACE performs best on most tasks (AIMNet best for $pK_a$). Full atom set improves performance; MLFF similarity outperforms LoCoHD. | Tables B.3-B.11, Fig. A.13 |

## B  ADDITIONAL FIGURES

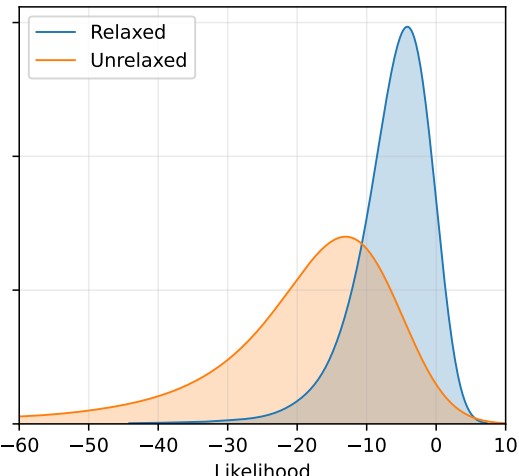

Figure A.7: **MACE-based likelihood captures structurally subtle distribution shifts in protein environments.** The histogram of likelihoods of CA MACE representations of 26000 protein environments curated from 225 protein structures in the test set, before (red) and after (blue) relaxing the protein structures with an Amber99 force-field (Hornak et al., 2006).

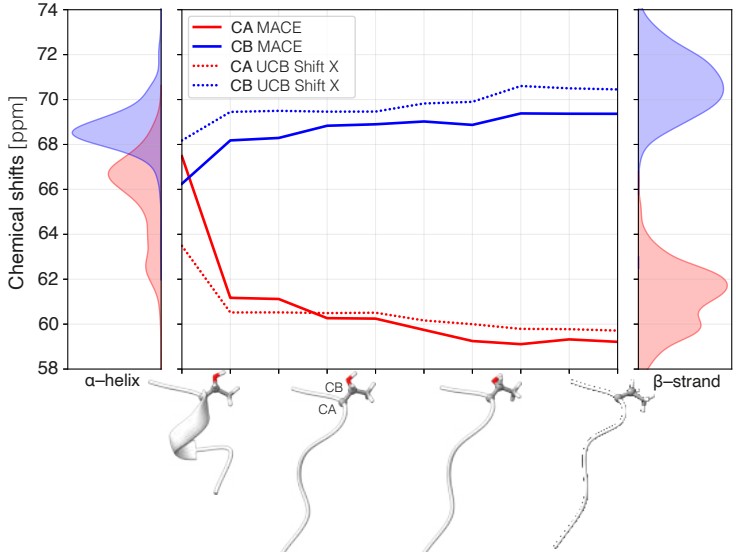

Figure A.8: **Simulated molecular dynamics trajectory of a helix unfolding into a strand.** Depicted are the chemical shifts of the CA and CB atoms in the middle of the helix as predicted by Mace and UCBShift2-X. Marginal plots show the experimentally-determined chemical shift distributions of the same atoms in helices (left) and strands (right). The known tendency of CA's to have smaller shifts in strands than in helices and the opposite CB tendency is clearly visible.

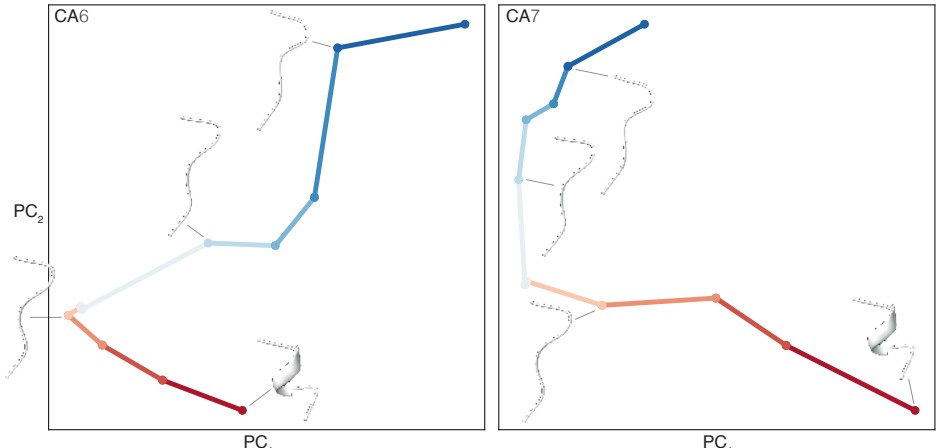

Figure A.9: **Structure of the MACE embedding space for the unfolding helix from Fig. A.8**. Depicted are Mace embeddings of the backbone CA atoms of residues 6 and 7 projected onto the first two principal components. Observe an essentially one-dimensional structure of the trajectory in the PCA space.

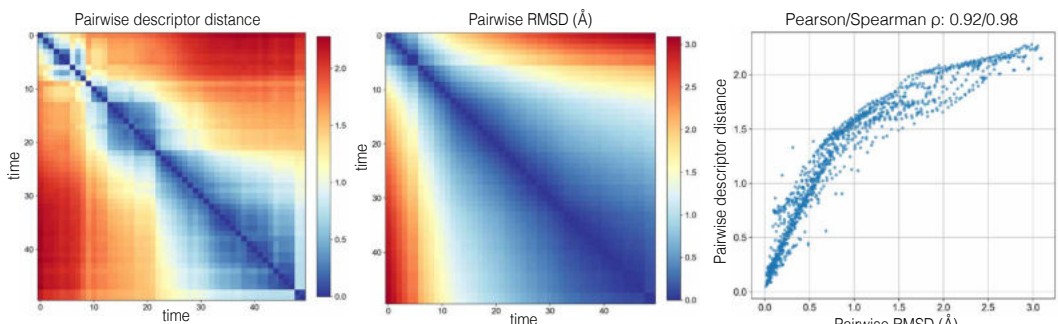

Figure A.10: **Pairwise descriptor distance is correlated with structural deviations.** (Left) Pairwise descriptor distances matrix for all frames along the unfolding MD trajectory. (Middle) Pairwise $C\alpha$ RMSD (Å) for the same set of structures. (Right) Relationship between descriptor distance and $C\alpha$ RMSD with Pearson/Spearman correlation coefficients of $\rho = 0.92/0.98$. The descriptor distance is the Euclidean distance between their vectorized $C\alpha$ feature representations.

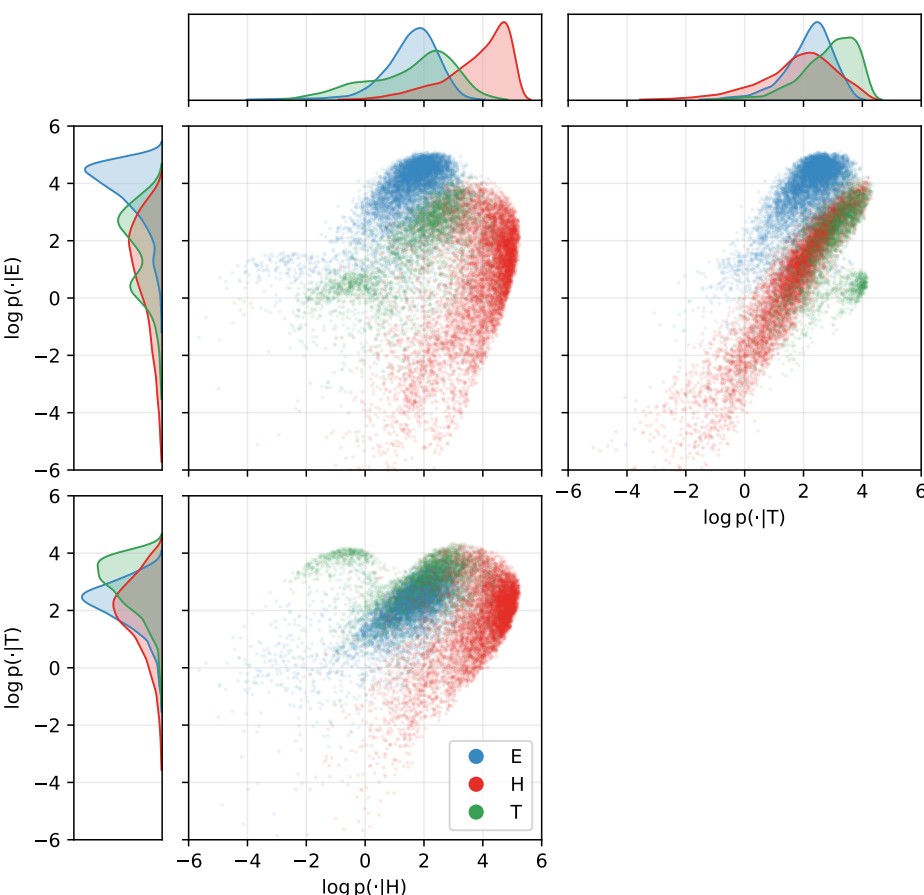

Figure A.11: **Estimated likelihoods of MACE embeddings of backbone CA atoms in different secondary structures.** The two-dimensional plots show projections of the log-likelihoods $(\log p(\mathbf{x}_i|E), \log p(\mathbf{x}_i|H), \log p(\mathbf{x}_i|T))$ of 8163 MACE embeddings $\mathbf{x}_i$ in strands (E), helices (H), and turns (T). The one-dimensional plots depict the KDE-estimated marginal distributions of each of the log-likelihoods.

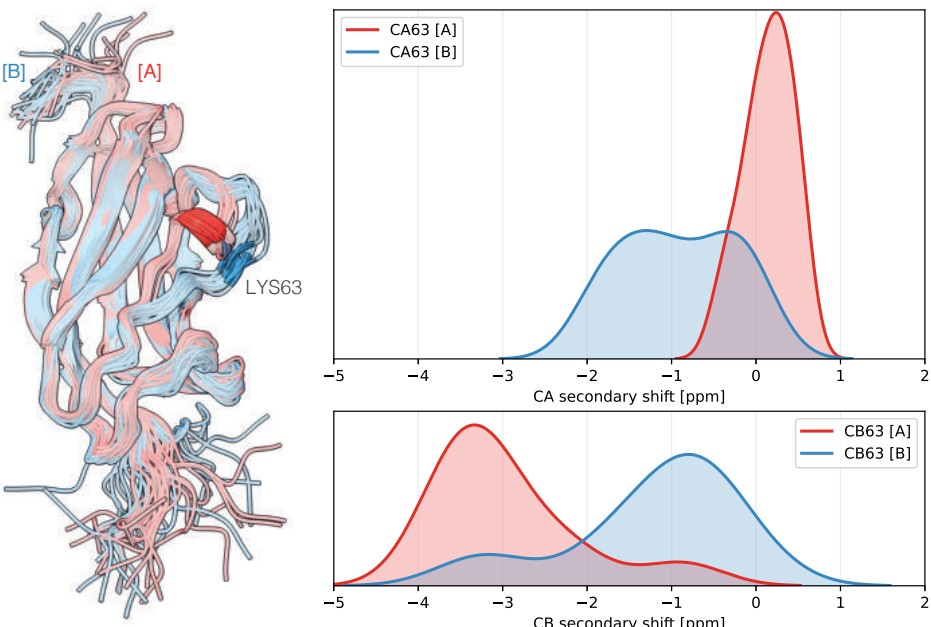

Figure A.12: **Distribution of the secondary chemical shifts of the two stable conformers of 4OLE.** Left: 100 ns MD simulations (Rosenberg et al., 2024) of the two conformers (marked as A (containing a helix) and B (containing a linearly structured loop) according to the original PDB annotation). Lysine 63 is highlighted. Right: secondary shift distributions of lysine 63 CA (top) and CB (bottom) atoms over the MD trajectory. Note that the helical conformer exhibits an expected excess in the CA secondary shift and a defect in the CB secondary shift.

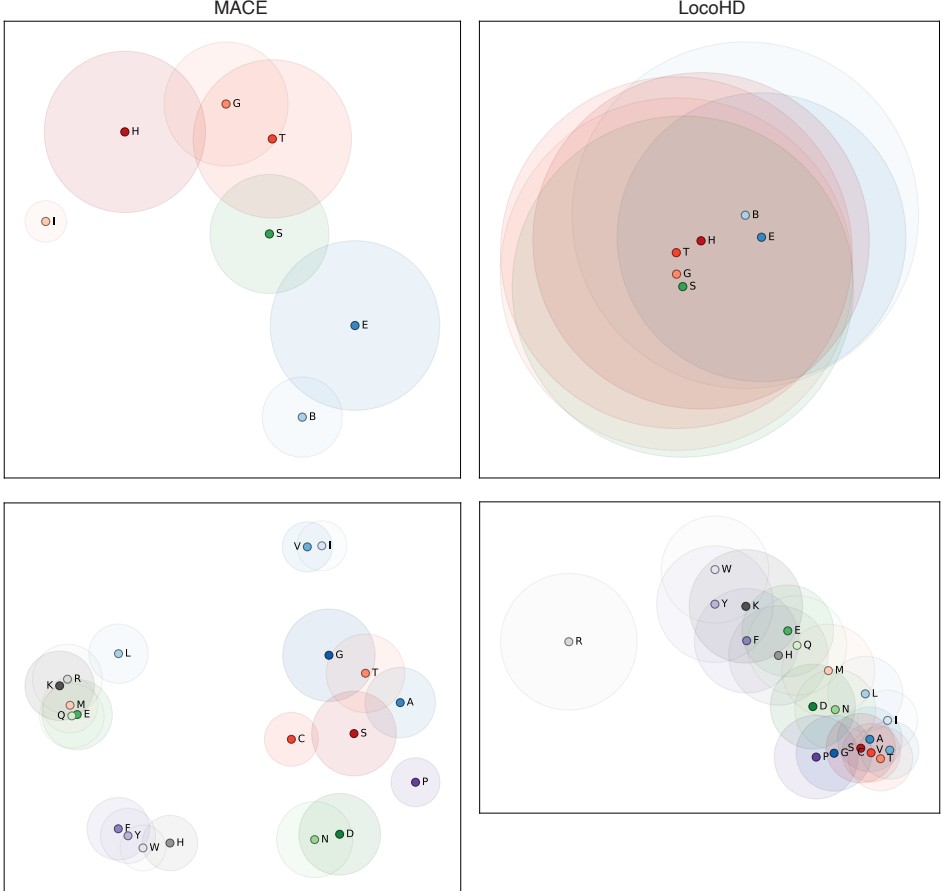

Figure A.13: **Similarity of local protein environments using MACE and LoCoHD.** In each plot, environments belonging to different class labels are represented as points while pairwise Euclidean distances approximate the dissimilarities as measured using the LoCoHD metric (Fazekas et al., 2024) and a metric between MACE likelihoods. Circles indicate each class variability as seen by the metric. Shown is conditioning by secondary structure (top row) and amino acid identity (bottom row). We conclude that MACE captures much better the similarities between related chemical and structural classes, even if LoCoHD has been explicitly designed for these tasks.

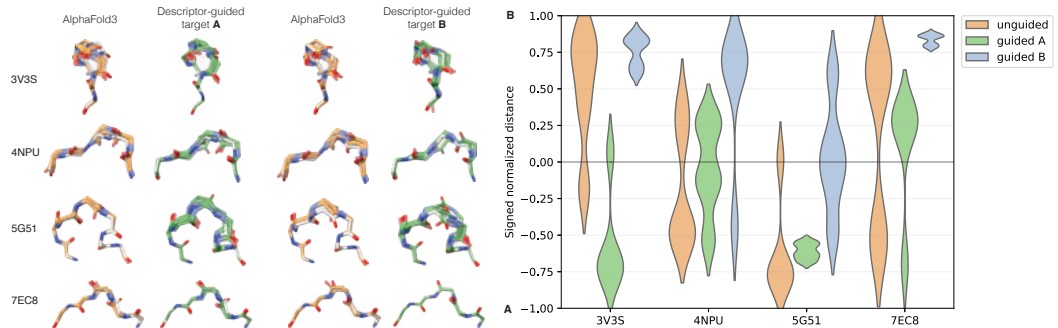

Figure A.14: **AlphaFold3 guided structures recovered by inverting MACE descriptors.** (A) The unguided AlphaFold3 structures (orange) of four proteins (`3V3S`, `4NPU`, `5G51`, and `7EC8`) with experimentally determined alternate backbone conformations are shown alongside guided predictions (green), obtained by inverting the corresponding conformation's MACE descriptor. All predictions are overlaid on the experimentally determined conformer (white) from the PDB (Burley et al., 2017). (B) The violin plots of the signed normalized distance distributions relative to conformations A ($-1$) and B (1) in generated structures for the same cases, comparing unguided (orange) and descriptor-guided predictions (green / blue).

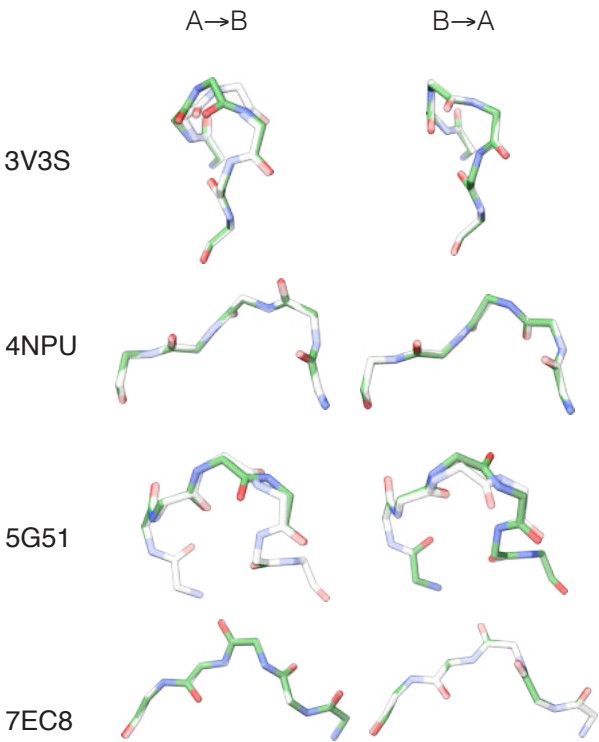

Figure A.15: **Inverting MACE descriptors by optimizing atomic coordinates.** Shown are the outcomes of optimizing atomic coordinates to match the local environment descriptors of an alternate conformation (altloc). For each protein, two optimization directions are displayed: starting from conformation A and matching the descriptors of conformation B (A→B), and starting from conformation B and matching the descriptors of conformation A (B→A). The proteins and altloc pairs used in this analysis are the same as those used for guidance in Fig. A.14. The reference conformation is shown in white, and the optimized conformation in green.

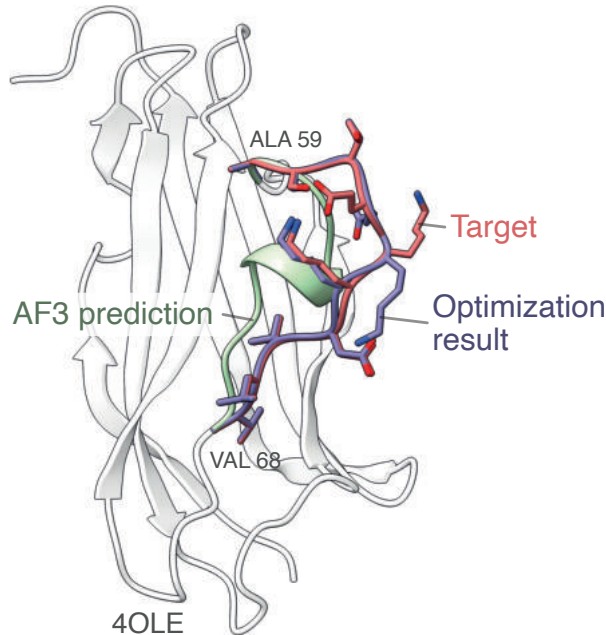

Figure A.16: **Inverting MACE descriptors by optimizing atomic coordinates** Shown in green is the structure of the protein 4OLE:B as predicted by AlphaFold3. The structure has an alternative conformation in the region 60 - 68, (shown in red is the crystallographic structure from the PDB), whose MACE environment descriptors were used to optimize the AlphaFold prediction. The solution of the optimization problem is depicted in pink.

## C  QUANTITATIVE RESULTS & TABLES

Table B.3: **Secondary structure prediction accuracy.** Reported are precision, recall, and F1 scores, each calculated with respect to the ground truth DSSP secondary structure classes. The best-performing model for each metric is indicated in **bold**.

| Metric | Class | LocoHD | ESM-Features + GCN | | | | MLFF-Features + GCN (Ours) | | | |
|--------|-------|--------|------------|-----|-------------|---------|-------|------|--------|--------|
| | | | ESM3 (Seq) | ISM | ESM3 (Struc) | ESMFold | Egret | MACE | OrbNet | AIMNet |
| | Weighted Mean | 76.947 | 88.175 | 92.467 | 94.881 | 93.333 | 95.685 | **95.694** | 92.500 | 94.716 |
| | Mean | 76.424 | 87.740 | 92.279 | 94.981 | 93.304 | **95.540** | 95.528 | 92.233 | 94.522 |
| Precision [%] | $\alpha$-helix | 80.493 | 90.303 | 93.065 | 96.103 | 94.431 | 96.466 | **96.657** | 93.900 | 95.762 |
| | $\beta$-strand | 75.623 | 91.124 | 95.555 | 95.252 | 95.757 | **96.248** | 95.969 | 93.749 | 95.494 |
| | Other | 73.158 | 81.793 | 88.216 | 92.419 | 88.941 | 93.905 | **93.958** | 89.049 | 92.309 |
| | Weighted Mean | 77.334 | 88.268 | 92.505 | 95.991 | 93.327 | **98.636** | 95.717 | 92.574 | 94.752 |
| | Mean | 75.532 | 87.637 | 92.071 | 95.799 | 92.961 | **98.508** | 95.355 | 91.977 | 94.335 |
| Recall [%] | $\alpha$-helix | 87.227 | 91.299 | 94.562 | 96.845 | 95.023 | **97.546** | 97.522 | 95.548 | 96.805 |
| | $\beta$-strand | 89.035 | 92.859 | 95.818 | 96.792 | 96.338 | **97.967** | 97.947 | 96.276 | 97.481 |
| | Other | 50.333 | 78.753 | 85.833 | 89.759 | 87.521 | 90.512 | **90.597** | 84.106 | 88.720 |
| | Weighted Mean | 76.323 | 88.210 | 92.479 | 95.433 | 93.311 | **98.633** | 95.693 | 92.509 | 94.720 |
| | Mean | 75.048 | 87.675 | 92.168 | 95.388 | 92.999 | **98.552** | 95.427 | 92.073 | 94.412 |
| F1 [%] | $\alpha$-helix | 83.725 | 90.798 | 93.807 | 96.473 | 94.726 | 97.003 | **97.087** | 94.717 | 96.281 |
| | $\beta$-strand | 81.783 | 91.983 | 95.687 | 96.016 | 96.047 | **97.100** | 96.948 | 94.996 | 96.477 |
| | Other | 59.636 | 80.244 | 87.008 | 91.069 | 88.225 | 92.177 | **92.247** | 86.507 | 90.479 |

Table B.4: **Amino acid prediction accuracy.** Reported are precision, recall, and F1 scores, each calculated with respect to the ground truth amino acid class. The best-performing model for each metric is indicated in **bold**. Unlike secondary structure prediction, per-class performance is not reported here due to the large number (20) of amino acid types.

| Metric | Class | LocoHD | ESM-Features + GCN | | | | MLFF-Features + GCN (Ours) | | | |
|---|---|---|---|---|---|---|---|---|---|---|
| | | | ESM3 (Seq) | ISM | ESM3 (Struc) | ESMFold | Egret | MACE | OrbNet | AIMNet |
| Precision [%] | Weighted Mean | 97.794 | 91.081 | 97.919 | 96.864 | 98.884 | **99.129** | 98.448 | 95.967 | 98.665 |
| | Mean | 98.012 | 89.923 | 96.891 | 96.828 | 97.821 | **98.758** | 98.031 | 95.090 | 98.407 |
| Recall [%] | Weighted Mean | 97.805 | 91.089 | 96.929 | 96.964 | 98.642 | **99.133** | 98.431 | 95.955 | 98.664 |
| | Mean | 97.699 | 89.484 | 95.922 | 96.803 | 97.823 | **98.932** | 98.262 | 95.072 | 98.457 |
| F1 [%] | Weighted Mean | 97.785 | 91.079 | 97.421 | 96.914 | 98.763 | **99.130** | 98.435 | 95.959 | 98.663 |
| | Mean | 97.839 | 89.686 | 96.404 | 96.815 | 97.822 | **98.842** | 98.137 | 95.078 | 98.431 |

Table B.5: **Backbone chemical shift prediction accuracy**. Reported is the mean absolute error (MAE), in ppm, relative to the experimentally measured chemical shift values from RefDB (Zhang et al., 2003) for different atoms. The best-performing model for each metric is indicated in **bold**.

| Atom | UCB Shift | LocoHD | Learned | ESM features + GCN | | | | | MLFF-features + GCN (Ours) | | | |
|---|---|---|---|---|---|---|---|---|---|---|---|---|
| | | | | ESM2 | ESM3 (Seq) | ISM | ESM3 (Struc) | ESMFold | MACE | Egret | Orb | AIMNet |
| HA | **0.165** | N/A | 0.405 | 0.247 | 0.235 | 0.220 | 0.193 | 0.203 | 0.180 | 0.176 | 0.200 | 0.201 |
| H | 0.324 | N/A | 0.528 | 0.402 | 0.393 | 0.377 | 0.600 | 0.365 | 0.300 | **0.295** | 0.333 | 0.325 |
| CA | 0.653 | 1.333 | 1.753 | 0.872 | 1.049 | 0.814 | 0.667 | 0.667 | **0.584** | 0.599 | 0.683 | 0.660 |
| N | 1.891 | 2.877 | 3.174 | 2.455 | 2.349 | 2.275 | 1.845 | 1.986 | **1.642** | 1.667 | 2.015 | 1.875 |
| CB | 0.758 | 1.140 | 1.326 | 0.923 | 1.092 | 0.884 | 0.758 | 0.775 | 0.716 | **0.705** | 0.792 | 0.780 |
| C | 0.827 | 1.297 | 1.517 | 1.011 | 1.038 | 0.923 | 0.842 | 0.876 | 0.744 | **0.743** | 0.847 | 0.824 |

Table B.6: **Side chain chemical shift prediction accuracy**. Reported is the mean absolute error (MAE), in ppm, relative to the experimentally measured chemical shift values from RefDB (Zhang et al., 2003) for different atoms. The best-performing model for each metric is indicated in **bold**.

| Atom | UCB Shift | LocoHD | MLFF-features + GCN (Ours) | | | |
|---|---|---|---|---|---|---|
| | | | MACE | Egret | Orb | AIMNet |
| CG | 0.595 | 0.647 | 0.567 | **0.563** | 0.638 | 0.611 |
| CD | 0.615 | 0.512 | 0.456 | **0.448** | 0.521 | 0.482 |
| CD2 | 1.116 | 1.195 | 1.035 | 1.085 | 1.201 | 1.749 |
| CG2 | 0.773 | 0.833 | **0.713** | 0.720 | 0.760 | 0.742 |
| CE | 0.519 | 2.295 | 0.447 | **0.436** | 0.451 | 0.446 |

Table B.7: **PropKa vs. MACE $pK_a$ comparison.** Per-residue $pK_a$ prediction errors for Glutamic acid and Aspartic acid from PropKa's training set. PropKa is evaluated in its native setting, using experimental $pK_a$ values derived from crystallographic PDB structures, whereas our MACE-based model was trained using simulated $pK_a$ values from AlphaFold2 structures. This comparison inherently favors PropKa, yet our model achieves lower error on more than half of the structures in PropKa's training set.

| PDB ID | Glutamic acid | | Aspartic acid | |
|---|---|---|---|---|
| | PropKa | Ours (MACE) | PropKa | Ours (MACE) |
| 1PGA:A | 0.413 | **0.294** | 0.362 | **0.292** |
| 1IGD:A | 0.489 | **0.182** | 0.352 | **0.274** |
| 1A2P:A | **0.525** | 1.046 | 0.426 | **0.403** |
| 4ICB:A | 0.579 | **0.546** | **0.299** | 1.018 |
| 1BEO:A | — | — | 0.730 | **0.511** |
| 4LZT:A | **0.629** | 0.665 | 0.488 | **0.404** |
| 3RN3:A | 0.548 | **0.462** | 0.423 | **0.412** |
| 2RN2:A | 0.274 | **0.185** | 0.683 | **0.585** |
| 2OVO:A | 0.377 | **0.376** | **0.180** | 1.571 |
| 1XNB:A | **1.001** | 1.270 | **0.436** | 1.020 |
| 135L:A | 0.470 | 1.700 | **0.480** | 1.043 |

Table B.8: **pK$_a$-ANI vs. MACE pK$_a$ performance.** Mean absolute pK$_a$ prediction error (MAE) for four amino acids using pK$_a$-ANI and our MACE-based model, evaluated on the PKAD-R dataset (Ancona et al., 2023), a curated version of the dataset on which pK$_a$-ANI was originally trained and tested. While pK$_a$-ANI is evaluated in its native setting, our model was trained on simulated pK$_a$ values derived from AlphaFold2 structures, making this comparison inherently favorable to pK$_a$-ANI. Nevertheless, our model outperforms pK$_a$-ANI on three of the four amino acids; for Histidine, we hypothesize that the poor performance is due to an insufficient number of training examples for that residue type.

| Amino acid | pKa-ANI | Ours (MACE) |
|---|---|---|
| Glutamic Acid | 0.696 | **0.695** |
| Aspartic Acid | 0.978 | **0.831** |
| Lysine | 0.648 | **0.360** |
| Histidine | **0.899** | 0.925 |

Table B.9: **Environment radius ablation.** Prediction error (ppm; lower is better) for backbone chemical shift prediction as a function of the local environment radius $r$. We compare $r = 4, 5, 6$ Å using MACE embeddings as input to the GCN. The best-performing radius (5 Å) is highlighted in bold.

| Atom | 4Å | 5Å | 6Å |
|---|---|---|---|
| HA | 0.194 | **0.180** | 0.191 |
| H | 0.320 | **0.300** | 0.318 |
| CA | 0.620 | **0.584** | 0.627 |
| N | 1.779 | **1.642** | 1.768 |
| CB | 0.758 | **0.716** | 0.754 |
| C | 0.798 | **0.744** | 0.804 |

Table B.10: **MLFF layer ablation.** Per-atom mean absolute error (MAE) in backbone chemical-shift prediction for different choices of descriptor layer. For MACE, L1 and L2 denote descriptors taken from the first and second message-passing layers, and L1+2 denotes their concatenation. For Orb, L5, L10, and L15 denote descriptors taken from the 5th, 10th, and 15th message-passing layers, respectively. For each atom type and model, the best-performing layer configuration is highlighted in bold.

| Atom | MACE L1 | MACE L2 | MACE L1+2 | Orb L5 | Orb L10 | Orb L15 |
|---|---|---|---|---|---|---|
| C | 0.797 | 0.755 | **0.744** | **0.836** | **0.836** | 0.847 |
| CA | 0.619 | 0.593 | **0.584** | 0.678 | **0.660** | 0.683 |
| CB | 0.756 | **0.705** | 0.716 | 0.778 | **0.777** | 0.792 |
| H | 0.318 | **0.300** | **0.300** | **0.330** | 0.331 | 0.333 |
| HA | 0.191 | **0.178** | 0.180 | 0.198 | **0.196** | 0.200 |
| N | 1.771 | 1.666 | **1.642** | 1.929 | **1.889** | 2.015 |

Table B.11: **Machine Learning Force Field (MLFF) metadata**. Listed are the model type, dimensionality, geometric properties of the representations, and training dataset for MLFFs in Section 2.

| Model Name | Dimension | Properties | Dataset |
|---|---|---|---|
| MACE large | 448 | $S_n$ & $O(3)$ invariant | QM9 (Ramakrishnan et al., 2014), MD22, ANI-1x |
| OrbNet-v2 | 256 | $S_n$ & $E(3)$ invariant | QM9, GDB13 (Cheng et al., 2019), DrugBank (Law et al., 2014) |
| AIMNet2 | 256 | $S_n$ & $E(3)$ invariant | ANI-1x (Smith et al., 2018) |
| Egret1 | 384 | $S_n$ & $O(3)$ invariant | QM9, MD22 (Chmiela et al., 2023) , ANI-1x, QM24 (Ruddigkeit et al., 2012) |

Table B.12: **Protein Secondary Structure Codes.** Summary of the standard single-letter codes used to represent protein secondary structure elements, as defined by the DSSP classification (Frishman & Argos, 1995).

| Symbol | Name |
|---|---|
| H | $\alpha$-helix |
| T | Turn |
| G | $3_{10}$-helix |
| I | $\pi$-helix |
| E | $\beta$-strand (extended) |
| B | Isolated $\beta$-hridge |
| S | Bend |
| – | Coil |

Table B.13: **Secondary structure prediction ablation.** Evaluation of secondary structure prediction accuracy when using a single Egret atom descriptor compared to using multiple Egret atom descriptors. The best-performing model for each metric is indicated in **bold**.

| Metric | Class | $\mathcal{A}_a = \{CA\}$ | $\mathcal{A}_a = \{C, CA, CB, N, H, HA\}$ |
|---|---|---|---|
| | Weighted mean | 94.822 | **95.685** |
| | Mean | 94.679 | **95.554** |
| Precision [%] | $\alpha$-helix | 95.809 | **96.466** |
| | $\beta$-strand | 95.232 | **96.248** |
| | Other | 92.997 | **93.905** |
| | Weighted mean | 94.853 | **98.636** |
| | Mean | 94.552 | **98.508** |
| Recall [%] | $\alpha$-helix | 96.759 | **97.546** |
| | $\beta$-strand | 97.971 | **99.467** |
| | Other | 88.925 | **90.512** |
| | Weighted mean | 94.817 | **98.633** |
| | Mean | 94.593 | **98.552** |
| F1 [%] | $\alpha$-helix | 96.282 | **97.003** |
| | $\beta$-strand | 96.582 | **97.103** |
| | Other | 90.915 | **92.177** |

Table B.14: **Amino acid prediction ablation.** Evaluation of amino acid prediction accuracy when using a single Egret atom descriptor compared to using multiple Egret atom descriptors. The best-performing model for each metric is indicated in **bold**.

| Metric | Class | $\mathcal{A}_a = \{CA\}$ | $\mathcal{A}_a = \{C, CA, CB, N, H, HA\}$ |
|---|---|---|---|
| Precision [%] | Weighted mean | 95.897 | **99.129** |
| | Mean | 94.327 | **98.758** |
| Recall [%] | Weighted mean | 95.912 | **99.133** |
| | Mean | 94.069 | **98.932** |
| F1 [%] | Weighted mean | 95.629 | **99.135** |
| | Mean | 93.765 | **98.842** |

Table B.15: Set of protein structures with multi-modal backbone conformations (altlocs) from Rosenberg et al. (2024) guided using MLFF descriptors.

| PDB ID | Residue range | Sequence | Resolution [Å] |
| --- | --- | --- | --- |
| 3V3S:B | $245 - 250$ | KAQERD | 1.90 |
| 7EC8:A | $187 - 190$ | DGGI | 1.35 |
| 5G51:A | $290 - 295$ | GSASDQ | 1.45 |
| 4NPU:B | $133 - 136$ | FEEI | 1.50 |

# D    EXTENDED BACKGROUND

**Chemical shifts.** The resonance frequency of a nuclear spin, $\nu$, depends directly on the splitting of its energy levels in a magnetic field. This is given as $\nu \propto \gamma B$, where $B$ is the strength of the magnetic field that the nucleus experiences and $\gamma$ is the gyromagnetic ratio of the nucleus, a characteristic of an atom's nucleus (e.g., $^1$H, $^{13}$C). In Nuclear Magnetic Resonance (NMR) experiments, $B$ is primarily determined by the externally applied magnetic field $B_0$, but at the level of an observed atom it is slightly perturbed by local magnetic fields $\Delta B$ induced by the surrounding electron cloud. These perturbations, which are typically several orders of magnitude weaker than $B_0$, cause small shifts in the resonance frequency – known as chemical shifts. The chemical shift reflects subtle variations in the local electronic environment and is therefore highly informative about molecular structure and composition. The chemical change of an atom is often reported in relative terms as $\delta = 10^6 \cdot \frac{\nu - \nu_0}{\nu_0}$, where $\nu$ is the observed resonance frequency and $\nu_0$ is the resonance frequency of the same type of nucleus in a reference compound – commonly the $^1$H signal of 2,2-dimethyl-2-silapentane-5-sulfonate (DSS) is used for the $^1$H chemical shift scale. Although dimensionless, chemical shifts $\delta$ are conventionally expressed in parts per million (ppm), hence the $10^6$ scaling factor in the definition.

The electronic environment that gives rise to chemical shifts is influenced by factors such as the nature of chemical bonds, the identities of neighboring atoms, and their three-dimensional spatial arrangement. Additionally, dihedral angles[3] play a significant role in modulating electron density around nuclei and thereby affect chemical shifts. Other factors that influence the chemical shift include hydrogen bonding interactions, ligand binding, or proximity to solvent molecules. Therefore, chemical shifts provide site-specific information about the local environment at the atomic-level. The chemical shift reports the time-averaged environment over all motions on time scales shorter than milliseconds. Chemical shifts are particularly valuable for inferring atomic connectivity, identifying functional groups, and detecting conformational changes, due to their sensitivity to fine variations in electronic structure (Claridge, 2016; Günther, 2013). Accurate chemical shift prediction plays a critical role in applications like automated NMR resonance assignment, molecular structure determination and validation, and the analysis of complex chemical systems (Wishart, 2011; Bermel et al., 2015). Despite this, reliable chemical shift prediction remains a challenge due to the sensitivity of chemical shifts to nuanced changes in molecular environment (Kuprov et al., 2007; Case, 2013).

**Acid dissociation constants (pK$_a$)** pK$_a$ is a thermodynamic quantity describing the tendency of a titratable group to donate a proton. When the solvent pH is lower than the pK$_a$, the group will be predominantly protonated; conversely, it becomes increasingly deprotonated at pH values above the pK$_a$. Each amino acid has a canonical pK$_a$ in isolation (e.g., Aspartate $\sim$3.9), but within proteins these values can shift significantly. Nearby charged or polar residues, hydrogen-bond partners, and desolvation effects (especially in buried regions) modulate proton affinity and can shift pK$_a$ by several pH units. Accurate prediction of these shifts is crucial for understanding enzymatic catalysis, pH-dependent conformational changes, and protein stability.

**Molecular Dynamics and Density Functional Theory.**    Molecular Dynamics (MD) simulations numerically integrate Newton's equations of motion to evolve the atomic positions of a system over time, requiring potential energies and forces at every timestep. When computed from quantum mechanics, these quantities can be obtained using Density Functional Theory (DFT), a first-principles electronic-structure method that solves the Kohn-Sham equations to deliver accurate total energies, forces, and electronic properties. However, the steep computational cost of DFT limits its direct application to small systems and short timescales. Machine learning force fields (MLFFs) address this limitation by providing near-DFT quality energies and forces at orders-of-magnitude lower cost, enabling quantum-accurate MD simulations of systems with tens of thousands of atoms over nanosecond timescales.

**ESM-based descriptors (Lin et al., 2023; Ouyang-Zhang et al., 2025; Hayes et al., 2025).** Evolutionary Scale Modeling (ESM) is a family of transformer-based protein language models trained on a large-scale dataset of protein amino acid sequences using masked language modeling, similar to BERT (Devlin et al., 2019) for natural language. Unlike traditional methods that rely on multiple sequence alignments (MSAs) or co-evolutionary profiles, ESM models learn context-aware representations directly from amino acid sequences, enabling them to capture biochemical,

---

[3]The torsion angle about the bond between atoms B and C in a four-atom segment A-B-C-D

structural, and evolutionary patterns within a protein sequence without explicit alignment. These representations have been used in two main ways. First, they underpin single-sequence 3D structure prediction in the ESMFold family of models (Lin et al., 2023). Second, later work augments these sequence-only models with structural supervision to obtain more structure-aware representations, for example in multimodal sequence-structure models such as ESM3 (Hayes et al., 2025) and in sequence-based protein language models that explicitly distill structural information into their embeddings (Ouyang-Zhang et al., 2025).

Given an input amino acid sequence $\mathbf{a}$ of length $L$, a pre-trained ESM model produces a $d$-dimensional representation for each amino acid. These embeddings capture the identity of each amino acid and its contextual dependencies within the sequence. In this work, we use per-residue representations from the ESM-2 model (output from the 33rd layer of the 2B parameter model) as local environment descriptors, leveraging the model's ability to capture the evolutionary and biochemical context for each residue.

**MACE and Egret descriptors (Batatia et al., 2022; Wagen et al., 2025).** Message passing neural networks (Gilmer et al., 2017; Schütt et al., 2017) typically exchange messages between pairs of nodes in a graph. In contrast, the MACE family of models (MACE, Egret) generalizes this framework by learning equivariant messages involving a system of $n$ (order parameter) nodes (or atoms) at a time. This enables richer geometric and chemical modeling by operating directly over higher-order interactions. Formally, for a given node at layer $t$ of the graph, MACE first learns the following $S_n$-invariant and $O(3)$-equivariant features between pairs of atoms:

$$\mathbf{A}^{(t)}_{i,kl_3m_3} = \sum_{l_1m_1,l_2m_2} C^{l_3m_3}_{l_1m_1,l_2m_2} \sum_{j\in\mathcal{N}(i)} R^{(t)}_{kl_1l_2l_3}(r_{ji})Y^{m_1}_{l_1}(\hat{\mathbf{r}}_{ij}) \sum_{\tilde{k}} W^{(t)}_{k\tilde{k}l_2} h^{(t)}_{j,\tilde{k}l_2m_2} \tag{2}$$

Here, $R$ is a learnable radial function that encodes the distance between atoms $i$ and $j$; $Y$ is the spherical harmonic function that encodes the unit vector from atom $i$ to atom $j$. The summation over the neighbors $\mathcal{N}(i)$ guarantees permutation invariance in the local aggregation. Additionally, $\mathbf{A}^{(t)}_{i,kl_3m_3}$ incorporates information from learned node embeddings at the previous layer using $\sum_{\tilde{k}} W^{(t)}_{k\tilde{k}l_2} h^{(t)}_{j,\tilde{k}l_2m_2}$. Lastly, $\sum_{l_1m_1,l_2m_2} C^{l_3m_3}_{l_1m_1,l_2m_2}$ (Clebsch-Gordan coefficients) ensure $O(3)$-equivariance is preserved when radial and spherical features are combined. The indices $l_1, l_2, l_3, m_1, m_2, m_3$ generalize equivariant messages across higher-dimensional irreducible representations of $O(3)$. It must be noted that $\mathbf{A}^{(t)}_{i,kl_3m_3}$ captures pairwise directional and distance-based interactions, forming the building blocks for higher-order representations. Formally,

$$\mathbf{B}^{(t)}_{i,\eta_\nu kLM} = \sum_{\mathbf{lm}} C^{LM}_{\eta_\nu,\mathbf{lm}} \prod_{\xi=1}^{\nu} \sum_{\tilde{k}} w^{(t)}_{k\tilde{k}l_\xi} \mathbf{A}^{(t)}_{i,\tilde{k}l_\xi m_\xi}, \mathbf{lm} = (l_1m_1, \ldots l_nm_n). \tag{3}$$

The above equation lifts the pairwise $S_n$ (permutation-) invariant and $O(3)$-equivariant features in $\mathbf{A}^{(t)}_{i,kl_3m_3}$ to an $n$-body representation while preserving the geometric properties. The high-order tensor $\mathbf{B}^{(t)}_{i,\eta_\nu kLM}$ forms the core MACE representation. The update and readout steps are similar to regular message passing frameworks. As a special case, we set $L = 0$ (scalar representation of $O(3)$) in Equation 3, we project the equivariant features to a invariant representation of $O(3)$ via scalar coupling. Hence,

$$\mathbf{B}^{(t)}_{i,\eta_\nu k,00} = \sum_{\mathbf{lm}} C^{00}_{\eta_\nu,\mathbf{lm}} \prod_{\xi=1}^{\nu} \sum_{\tilde{k}} w^{(t)}_{k\tilde{k}l_\xi} \mathbf{A}^{(t)}_{i,\tilde{k}l_\xi m_\xi}, \mathbf{lm} = (l_1m_1, \ldots l_nm_n). \tag{4}$$

Here $C^{00}_{\eta_n,\mathbf{lm}}$ would enforce scalar coupling of lower-order equivariant components. In this work, we will use descriptors (448 dimensional) that are invariant to both $S_n$ and $O(3)$ transformations by setting $L = 0$.

**AIMNet2-based descriptors (Zubatyuk et al., 2019).** AIMNet constructs Atomic Feature Vectors (AFVs) by iteratively embedding the local atomic environments and updating atomic representations using geometric information. Given the atomic coordinates $\mathbf{R}$ and atomic number $Z$, an initial atomic feature vector $\mathbf{A}^{(0)}_i$ is derived for atom $i$. The initial embeddings are parametrized as learnable vectors that depend on atomic number $Z_i$ of each atom.

The local environment of atom $i$ is encoded using symmetry functions that capture the spatial arrangement and types of neighboring atoms. This produces an atomic environment vector (AEV).

$$\mathbf{G}_i^{(t)} = \text{AEV}(\{\mathbf{R}_i - \mathbf{R}_j, Z_j\}_{j \in \mathcal{N}(i)}). \tag{5}$$

An AEV is constructed to be invariant to permutations ($S_n$) and 3D translations and rotations ($E(3)$). This is typically via a message-passing framework. Next, the geometry and atom-type information are fused by computing an interaction descriptor

$$\mathbf{f}_i^{(t)} = \text{MLP}\left(\sum_{j \in \mathcal{N}_i} \mathbf{G}_{ij}^{(t)} \otimes \mathbf{A}_j^{(t)}\right) \tag{6}$$

This vector encodes the influence of neighboring atoms on atom $i$ in a way that is independent of the number of chemical species. Lastly, the atomistic feature vectors $\mathbf{A}_j^{(t)}$ and $\mathbf{f}_i$ are combined to form an updated atomistic feature vector

$$\mathbf{A}_i^{(t+1)} = U(\mathbf{A}_i^{(t)}, \mathbf{f}_i^{(t)}). \tag{7}$$

This process is repeated over multiple iterations, enabling information to propagate beyond the local neighborhood and capture non-local effects such as charge redistribution and polarization. After several refinement steps, AIMNet produces AFVs of fixed dimensionality (256), which are used to predict various molecular and atomic properties. Training is performed in a multitask setting, where the network jointly learns to predict a variety of molecular and atomic properties – including energies, forces, partial charges, and dipole moments – using shared representations to improve generalization. The model is trained on large datasets of quantum mechanical calculations, with regularization techniques to ensure stability and prevent overfitting.

**OrbNet-based descriptors (Qiao et al., 2020).** In similar spirit to semi-empirical molecular force-fields, OrbNet constructs feature vectors by leveraging symmetry-adapted atomic orbitals (AO), which capture local electronic structure information obtained from quantum calculations while respecting the symmetries inherent in molecular systems. Using mean-field quantum mechanics, OrbNet derives overlap matrices $\mathbf{S}$, Fock matrix $\mathbf{F}$, density matrix $\mathbf{D}$, and the Hamiltonian $\mathbf{H}$. Each matrix element $O^{\nu\mu}$ corresponds to interactions between atomic orbitals $\nu$ and $\mu$, and blocks $\mathbf{O}_{ij}$ represent interactions between atoms $i$ and $j$.

These matrices produce features that are $S_n$ invariant and $E(3)$ invariant by projecting these orbital block matrices into a symmetry-adapted basis. The atom-wise feature vector is then constructed by aggregating contributions from its neighboring orbital interactions

$$\mathbf{f}_i = \sum_j \text{MLP}(\mathbf{O}_{i,j}). \tag{8}$$

In OrbNet, the atomic structure is represented as a graph where nodes correspond to an atom (initialized using $\mathbf{f}_i$) and edges model interatomic relationships. The model employs the following message passing scheme

$$\mathbf{h}_i^{(t+1)} = U^{(t)}\left(\mathbf{h}_i^{(t)}, \sum_{j \in \mathcal{N}(i)} M^{(t)}(\mathbf{h}_i^{(t)}, \mathbf{h}_j^{(t)}, \mathbf{O}_{ij})\right), \tag{9}$$

where $\mathbf{h}_i^{(t)}$ is hidden feature of atom $i$ at layer $t$, $\mathbf{O}_{ij}$ encodes the orbital-based edge features derived from orbital feature matrices, $M^{(t)}$ is the message function, and $U$ is the update function for node embeddings. After multiple layers of message passing, the final atomic features $\mathbf{f}_i \in \mathbb{R}^{256}$ serves as a descriptor of an atom's local environment. This formulation enables OrbNet to encode both local chemical environments and long-range quantum effects efficiently. In contrast to models like MACE, which explicitly encode geometric equivariance and higher-order correlations, OrbNet uses pairwise orbital-based features and rely on message passing to capture more complex interactions.

**LoCoHD** (Fazekas et al., 2024) is a method for quantifying chemical and structural differences between protein environments. Unlike alignment-based or purely geometric measures, LoCoHD characterizes each local environment as *a distribution of chemical "primitive types"* such as atom

types or residue centroids within a specified radius. The similarity between two environments is then measured using the Hellinger distance between their respective distributions. In this paper, to test the effectiveness of LoCoHD descriptors in representing local environments, we construct *LocoHD embeddings* using its representation of a local environment while computing the similarity metric. Each embedding is computed by aggregating statistics of the primitive types, weighted by their distance from the central residue. This results in a purely structural and chemical descriptor of the local environment.

## E    DATA CURATION

The dataset used within the scope of this paper was sourced from RefDB (Zhang et al., 2003) which is a subset of the Biomolecular Magnetic Resonance Databank (BMRB) (Hoch et al., 2023). RefDB is a curated list of BMRB entries with calibrated chemical shifts. We select monomers from the RefDB and perform sequence redundancy filtering with `mmseqs` via,

```
mmseqs easy-cluster sequences.fasta --min-seq-id 0.5 -c 0.8
--cov-mode 5
```

which results in sequence clusters. We then split the clusters to define train and test sets. Our final dataset consists of $1048$ BMRB entries in RefDB ($225$ test + $823$ train), from where we collected the amino acid sequences, and the experimental chemical shifts.

**Why use AlphaFold structures.**    Instead of using Protein Data Bank (PDB) structures, we predict 3D structures for all entries with AlphaFold2 (Jumper et al., 2021) via OpenFold (Ahdritz et al., 2024). Using experimental crystal structures would introduce several limitations:

- **Poor generalization.** X-ray crystallography covers only a small fraction of the proteome, meaning a model trained on PDB-derived environments would not generalize to most sequences without available structures.

- **Different experimental conditions.** Crystal structures and NMR represent different physical states (solid vs. solution), leading to systematic differences in local geometry that can affect chemical shifts.

- **Sequence mismatches and ambiguity.** We observed $\sim15.8\%$ sequence misalignment between BMRB entries and corresponding PDB structures. A single BMRB ID can map to multiple PDB entries (e.g., BMRB 15064 $\rightarrow$ 16 PDBs), making it unclear which structure should be used as reference.

AlphaFold alleviates these issues by providing systematic and consistent structural predictions for every sequence, including those with no experimental structure, enabling uniform training and evaluation.

Local protein environments are extracted from each AlphaFold structure as described in Algorithm 1.

Random-coil chemical shifts used in our predictor are obtained from UCBShift2 (Ptaszek et al., 2024).

Due to the limited availability of high-quality experimental training data, our $pK_a$ prediction models are regressed over synthetic values determined by solving the Poisson-Boltzmann equation via the method of Reis et al. (2020).

## F    ENVIRONMENT CONSTRUCTION AND MLFF EMBEDDING

This section elaborates on the amino acid centric environment construction algorithm discussed in Section 3.

---

**Algorithm 1** Amino Acid Centric Environment Extraction

---

**Require:** Protein dataset $\mathcal{D}$, radius $r_{\max} = 5\text{Å}$
1: $\mathcal{E} \leftarrow \emptyset$
2: **for all** structure $\mathcal{X} \in \mathcal{D}$ **do** $\qquad\qquad\qquad\qquad\qquad\qquad\qquad\qquad\quad \triangleright \mathcal{X} \in \mathbb{R}^{N_{\text{atoms}} \times 3}$
3: $\quad$ **for all** amino acid $a \in \mathcal{X}$ **do**
4: $\quad\quad$ **for all** atom $\mathbf{x}_i \in a$ **do** $\qquad\qquad\qquad\qquad\qquad\qquad\qquad\quad \triangleright \mathbf{x}_i \in \mathbb{R}^3$
5: $\quad\quad\quad$ $\mathcal{X}_a \leftarrow \{a\}$
6: $\quad\quad\quad$ **for all** amino acid $a' \in \mathcal{X}$ **do**
7: $\quad\quad\quad\quad$ **if** $\exists \mathbf{x}_j \in a'$ s.t. $\|\mathbf{x}_i - \mathbf{x}_j\| \leq r_{\max}$ **then**
8: $\quad\quad\quad\quad\quad$ $\mathcal{X}_a \leftarrow \mathcal{X}_a \cup \{a'\}$ $\qquad\qquad\qquad\qquad\qquad \triangleright$ All atoms in $a'$
9: $\quad\quad\quad\quad$ **end if**
10: $\quad\quad\quad$ **end for**
11: $\quad\quad\quad$ $\mathcal{E} \leftarrow \mathcal{E} \cup \mathcal{X}_a$
12: $\quad\quad$ **end for**
13: $\quad$ **end for**
14: **end for**
15: **return** $\mathcal{E}$

---

## F.1 RADIUS ABLATION

To select the radius used for environment construction, we performed an ablation study over $r \in \{4, 5, 6\}$ Å using MACE (Kovács et al., 2023) embeddings for backbone chemical shift prediction. MACE is trained with a cut-off of 6 Å, so larger radii are not meaningful. The results (Table B.9) show that a radius of 5 Å performs best. We hypothesize that this radius is large enough to capture the relevant NMR chemical-shift environment while still keeping most residue neighborhoods as complete as possible under the 6 Å cut-off, thereby providing a good balance between coverage and truncation.

## F.2 MLFF LAYER COMPARISON

This ablation study examines how selecting different hidden layers as MLFF embeddings affects performance for MACE and Orb. MACE has two hidden message-passing layers; throughout this work, the MACE descriptor is defined as the concatenation of the invariant components of their hidden states, matching the default behavior of the `mace.get_descriptors` function. For all other MLFF models, the descriptor corresponds to the hidden state of the final message passing layer. Orb has fifteen hidden message-passing layers, so in this case the descriptor is given by the hidden state of the 15th layer.

Table B.10 summarizes the per-atom mean absolute errors for all evaluated layers on the chemical-shift prediction task. For MACE, the concatenation of its two hidden layers yields the best performance for most atom types, while for Orb the 10th hidden layer performs best in most cases. Nevertheless, throughout this work we use the 15th Orb layer to maintain a consistent configuration across MLFFs with many layers, since concatenating all hidden states would result in impractically high-dimensional descriptors.

## G PREDICTING AMINO ACID, SECONDARY STRUCTURE, AND PK$_\text{A}$

In what follows, we provide additional details for experiments and results described in Section 4.

### G.1 SECONDARY STRUCTURE AND AMINO ACID PREDICTION

**Target variables.** In these tasks, the target space of the target y is categorical variable.

- In **amino acid prediction**, the target variable y holds one of the following categories: $\{A, C, D, E, F, G, H, I, K, L, M, N, P, Q, R, S, T, V, W, Y\}$, each representing a standard twenty amino acids (cf. (Kawashima & Kanehisa, 2000) for further explanation).

- In **secondary structure prediction**, the target variable $\mathbf{y}$ can assume one of the following categories: $\{H, E, O\}$, where $H$ denotes an $\alpha$-helix, $E$ denotes a $\beta$-strand, and $O$ corresponds to all other secondary structure types (see Table B.12). Although more secondary structure types exist, we are primarily interested in $\alpha$-helices and $\beta$-strands due to their structural prominence.

**Loss function.**     To train models for these tasks, we employ a *cross entropy loss* written as

$$\mathcal{L}(\hat{y}_i, \mathbf{y}_i) = -\sum_{c=1}^{|\mathbf{y}|} y_{ic} \log \left( \frac{\exp(\hat{y}_{ic})}{\sum_{j=1}^{|\mathbf{y}|} \exp(\hat{y}_{ij})} \right), \tag{10}$$

where $\hat{y}_i$ is the *predicted logit* for each class and $\mathbf{y}_i$ is the *one-hot encoding* of the true class label for sample $i$.

**Evaluation metrics.**     To evaluate the model's performance, we use the following metrics.

- *Precision*: For a given class $i$, precision is defined as

$$\text{Precision}_i = \frac{\text{True Positive}_i}{\text{True Positive}_i + \text{False Positive}_i}$$

- *Recall*: For a given class $i$, recall is defined as

$$\text{Recall}_i = \frac{\text{True Positive}_i}{\text{True Positive}_i + \text{False Negative}_i}$$

- *F1 score*: For a given class $i$, the F1 score is the harmonic mean of its Precision and Recall:

$$\text{F1}_i = 2 \cdot \frac{\text{Precision}_i \cdot \text{Recall}_i}{\text{Precision}_i + \text{Recall}_i}$$

For secondary structure prediction we report the precision, recall, and F1 score for each class in $\mathbf{y}$. In addition, we report an unweighted average and a weighted average (mean weighted by number of instances in each class) for the respective tasks.

**Ablation study.**     We conduct an ablation study on the set of atoms $\mathcal{A}_a$ to evaluate the effect of including descriptors from multiple atoms within amino acid $a$ on secondary structure and amino acid prediction performance. Specifically, two configurations are compared: one where $\mathcal{A}_a = \{\text{C, CA, CB, N, H, HA}\}$, and a reduced version $\mathcal{A}_a = \{\text{CA}\}$. For this ablation, identical models are trained using Egret descriptors, varying only the atom set. Egret descriptors are used because as shown in Tables B.3 and B.4, they consistently outperform other atom-level descriptor types. The ablation results for secondary structure prediction are presented in Table B.13 and for amino acid prediction in Table B.14. Clearly, incorporating a larger set of atoms in $\mathcal{A}_a$ leads to a substantial improvement in prediction accuracy.

## G.2     Acid dissociation constant (pK$_\text{A}$) prediction

**Loss function.**     We treat $pK_\text{a}$ prediction as a regression problem. To train pKa regressors, we train a graph neural network to minimize the mean absolute error (MAE) as follows,

$$\mathcal{L}_1 = \frac{1}{N} \sum_{i=1}^{N} |y_i - \hat{y}_i|, \tag{11}$$

where $N$ is the number of samples in the dataset, $y_i^*$ is the *groundtruth* pK$_\text{a}$ value of a protonated site in a protein obtained by solving the Poisson-Boltzmann equation from Reis et al. (2020), and $\hat{y}_i$ is the *predicted* pK$_\text{a}$.

**Comparison on experimental data.** To quantitatively assess the practical performance of our approach, we compare against $pK_a$-ANI Gokcan & Isayev (2022) and PROPKA Olsson et al. (2011) on the benchmark summarized in Table B.7. Both reference methods were originally trained on experimental $pK_a$ values derived from PDB structures, in contrast our models were trained on simulated values and predominantly on AlphaFold2 structures.

For $pK_a$-ANI, we evaluate on the PKAD-R (Ancona et al., 2023) dataset, a curated and updated version of the original PKAD dataset used for their training. Because no explicit train–test split was reported, we benchmark both models on all entries in PKAD-R. For PropKa, we evaluate on their training set, for which exact predictions were provided. The authors of PropKa also mention evaluation on a set of 201 proteins assembled from three previously published datasets, but they do not specify the corresponding PDB identifiers or make their predictions available, preventing a direct comparison on that benchmark.

These comparisons are conservative with respect to our method, because the baselines are trained on experimental PDB-based data whereas our models are trained on simulated labels and AlphaFold structures. Nevertheless, our model outperforms PropKa on more than half of the entries and matches or surpasses $pK_a$-ANI on 3 out of 4 residue types. For the remaining residue type (Histidine), we hypothesize that the weaker performance is due to the substantially smaller number of available training examples.

### G.3 MODEL ARCHITECTURE

Across all three tasks, we used a consistent model architecture with little changes in hyperparameters. To incorporate the spatial and relation information between atomistic features, we constructed *a fully connected graph between the atoms of the same residue*. We propose to use a graph convolution network (GCN) (Kipf & Welling, 2016) to predict the specific class type. The update rule at layer $l \in \mathbb{N}$ is defined as,

$$\mathbf{X}^{(l+1)} = \sigma(\bar{\mathbf{A}}\mathbf{X}^{(l)}\mathbf{W}^{(l)}) \tag{12}$$

Here, $\bar{\mathbf{A}} = (\mathbf{I} + \mathbf{A})$ is the normalized adjacency matrix that serves as a low-pass filter when aggregating information over the neighbors. $\mathbf{A}$ is the graph's adjacency matrix, and $\mathbf{I}$ is an identity matrix. Also, $\mathbf{W}^{(l)}$ is the learnable weight matrix to transform the features at layer $l$, $\mathbf{X}$ is the node feature matrix defined for layer $l$ of the GNN. We used SiLU (Sigmoid Linear Unit) (Elfwing et al., 2018) as the non-linear activation function and we applied layer normalization after each graph convolution operation to stabilize training. The operation in Eq. 12 is repeated for $L = 10$ layers. The initial input $\mathbf{X}^{(0)}$ consists of the atomistic features for each atom. At the final layer ($l = L - 1$), in the case of classification, the network outputs a vector of shape $|\mathrm{y}|$ representing class-wise logits, and the network outputs a scalar for regression tasks. Lastly, we use dropout (Srivastava et al., 2014) for every intermediate layer to prevent overfitting. Below is the table of hyper-parameters used for these experiments.

- $\mathcal{A}_a = \{\mathrm{C}, \mathrm{CA}, \mathrm{CB}, \mathrm{N}, \mathrm{H}, \mathrm{HA}\}$
- Batch size: 256
- Number of Epochs: 800
- Optimizer
  - Type: Adam (Kingma & Ba, 2015)
  - Learning Rate: $1 \times 10^{-4}$
- Learning Rate Scheduler
  - Type: Step Learning Rate
  - Decay Factor ($\gamma$): 0.5
  - Step Size: 10
- Number of GCN layers: 10
- Dropout probability: 0.3
- Hidden dimension of GCN (per layer): 128

In addition, we employed gradient clipping with a maximum norm of 5, an exponential moving average (EMA) weighted model with a decay rate of 0.999, and a StepLR learning rate scheduler incorporating 7500 warm-up steps and a step size of 10.

## H    USING LIKELIHOODS TO DEFINE ENVIRONMENT SIMILARITY MEASURE

We employ the conditional likelihoods defined in Section 5 to define a similarity metric to compare protein environments.

### H.1    COMPUTATION DETAILS

All experiments were performed on a single NVIDIA H100 GPU. Individual runs required one hour to complete, with a maximum GPU memory usage of approximately 15 GB.

### H.2    SIMILARITY METRIC

We analyzed a large collection of protein chemical environments by comparing them using MACE embeddings. For each attribute – such as amino acid type or secondary structure – we estimated a likelihood around each embedding via kernel density estimation with a fixed bandwidth as described in Section 5. These likelihoods were converted into distances using a scaled negative exponential, where higher similarity corresponded to smaller distances. Since each pairwise distance arises from a distribution – based on multiple environments belonging to the same attribute – we applied stochastic multidimensional scaling (MDS) (Rosenberg et al., 2022; Boyarski & Bronstein, 2021), which takes into account both the mean and standard deviation of these distances. The resulting embeddings revealed meaningful clustering that not only separated different amino acid types and secondary structures but also captured finer relationships within them. Chemically or structurally related amino acids and secondary structures exhibited higher similarity, demonstrating that the MACE-based comparison effectively reflects the underlying chemical and structural properties. Additional details and visualizations are provided in Figure A.13.

## I    CHEMICAL SHIFT PREDICTION

### I.1    EXTENDED PHYSICS-GROUNDED CHEMICAL SHIFT RESULTS

In Section 6 we presented one representative case study to examine whether the shift predictor correctly captures local structural influences on chemical shifts. Here, we provide two additional case studies to further validate these findings. Both probe distinct types of conformational variation and demonstrate that the proposed predictor consistently produces physically meaningful chemical shift trends.

**Helix unfolding into a strand.** Chemical shifts of CA and CB have distinct behaviors in helices and strands. We perform an MD simulation of a 7-residue peptide unfolding from a helix to a strand. Predicted chemical shifts along the trajectory are depicted in Fig. A.8. We note that our method correctly reproduces the experimentally observed chemical shift distributions for CA and CB atoms. It clearly captures the characteristic pattern where CA shifts decrease in strands compared to helices, while CB shifts exhibit the opposite trend.

**Alternative conformations.** We evaluate our shift predictor over a 100 ns MD simulation of the protein 4OLE:B from Rosenberg et al. (2024), which features two stable conformers: conformer A containing a helix and conformer B with a linearly structured loop (Fig. A.12). Our method captures distinct secondary chemical shift distributions for lysine 63 CA and CB atoms. The helical conformer exhibits the expected increase in CA secondary shift and decrease in CB secondary shift.

### I.2    DETAILS OF THE CHEMICAL SHIFT PREDICTION EXPERIMENTS

**Target variable.**    The goal of the chemical shift prediction is to predict the calibrated chemical shift $y$ of an atom of interest given the environment. We train our models to predict the *secondary shift*, i.e. the difference in the experimental and random-coil chemical shifts, given the sequence.

**Loss function.** We train our shift prediction models to minimize the mean absolute shift prediction error, defined as

$$\mathcal{L}_{\text{L1}} = \frac{1}{N} \sum_{i=1}^{N} |y_i - \hat{y}_i|, \tag{13}$$

where $N$ is the number of data points, $y_i$ and $\hat{y}_i$ are the groundtruth and predicted chemical shifts, respectively.

**Evaluation metrics.** To quantitatively evaluate the accuracy of the model, we report the mean absolute error measured on the test set.

### I.3 MODEL ARCHITECTURE

We used the same architecture as before (Appendix G.3) with slightly different hyperparameters as described below.

- $\mathcal{A}_a = \{\mathbf{b}\}$, where $\mathbf{b}$ is the atom type for which we are predicting the chemical shift value.
- Batch size: 500
- Number of Epochs: 2000
- Optimizer
  - Type: Adam (Kingma & Ba, 2015)
  - Learning Rate: $1 \times 10^{-4}$
- Learning Rate Scheduler
  - Type: Step Learning Rate
  - Decay Factor ($\gamma$): 0.5
  - Step Size: 10
- Number of GCN layers: 5
- Dropout probability: 0.3
- Hidden dimension of GCN (per layer): 256

In addition, we employed gradient clipping with a maximum norm of 5, an exponential moving average (EMA) with a decay rate of 0.999, and a StepLR learning rate scheduler incorporating 7500 warm-up steps and a step size of 10.

### I.4 COMPUTATION DETAILS

All experiments were performed on a single NVIDIA H100 GPU. Individual runs required three hours to complete, with a maximum GPU memory usage of approximately 15 GB.

## J EXTENDED MLFF INTERPRETATIONS

### J.1 HELIX UNFOLDING

As a complement to the main-text case study in Section 7, we analyze the behavior of MLFF embeddings during a helix-to-strand transition. Using MD simulations of a 7-residue peptide (Fig. A.9), we perform PCA on the MACE embeddings of CA atoms in residues 6 and 7, and observe that they follow smooth, non-intersecting trajectories. These trajectories reflect the gradual unfolding process, indicating that the descriptor space encodes continuous and interpretable structural transitions.

To further quantify how the learned descriptors relate to structural changes, we compare pairwise distances in descriptor space with backbone $C\alpha$ RMSDs along the same trajectory (Fig. A.10). For each pair of MD frames, we compute the Euclidean distance between the vectorized MACE embeddings of all $C\alpha$ atoms in the protein and plot this as a distance matrix alongside the corresponding RMSD

matrix, as well as a scatter plot of descriptor distance versus RMSD. The strong Pearson/Spearman correlations ($\rho = 0.92/0.98$) and the similar block structure of the two matrices show that conformations that are close in 3D space are also close in descriptor space, while structurally dissimilar states are well separated. This approximate isometry implies that the MLFF descriptors preserve the geometry of the unfolding pathway, thereby supporting their interpretability: distances and trajectories in embedding space can be directly related to physically meaningful structural deviations.

## J.2 INVERTING MACE EMBEDDINGS

In this section, we explore whether MACE descriptors can be inverted to recover protein structures and distinguish between alternate conformations (altlocs). We treat MACE features not only as structural representations but also as potential tools to decode conformational states. We present two complementary approaches: (*i*) direct optimization of atom positions to fit MACE embeddings, and (*ii*) in line with Maddipatla et al. (2025), guiding AlphaFold3's reverse diffusion process using MACE descriptors.

### J.2.1 DIRECT OPTIMIZATION OF ATOMIC COORDINATES

We first investigate whether MACE features can serve as latent representations from which protein conformers can be reconstructed. As a test case for descriptor-based structure inversion, we examined the protein `4OLE:B`, which contains a nine–residue segment (residues 60–68) modeled as two distinct alternate conformations. In addition to this example, we applied the same analysis to the proteins listed in Table B.15: `3V3S:B`, `7EC8:A`, `5G51:A`, and `4NPU:B`. Each of these structures contains a short region annotated with alternate conformations, typically involving subtle backbone displacements rather than large-scale structural rearrangements. For every protein, we optimized the atomic coordinates in the altloc region so that the local MACE descriptors of one conformation match those of the other, enabling us to assess how well the descriptors capture and recover experimentally observed conformational differences. The resulting optimized structures are shown in Fig. A.16 and Fig. A.15.

The optimization loss function quantifies the discrepancy between the descriptors of the current and target structures and is computed for each atom within the optimized region. In particular, we define each local atomic environment as all atoms within a 5 Å radius around the atom of interest. The resulting descriptors capture both geometric and chemical context at multiple equivariant orders. The loss is composed of three terms:

$$\mathcal{L} = \alpha \left\| \mathbf{B}_{i,\eta_\nu k,00} - \mathbf{B}^{\text{target}}_{i,\eta_\nu k,00} \right\|_2 + \beta \frac{1}{N} \sum_{i=1}^{N} \left\| \mathbf{B}^i_{i,\eta_\nu k,1M} - \mathbf{B}^{\text{target},i}_{i,\eta_\nu k,1M} \right\|_2$$
$$+ \gamma \sum_i \left\| \mathbf{B}^i_{i,\eta_\nu k,2M} - \mathbf{B}^{\text{target},i}_{i,\eta_\nu k,2M} \right\|_F,$$

where $\mathbf{B}^i_{i,\eta_\nu k,LM}$ and $\mathbf{B}^{\text{target}}_{i,\eta_\nu k,LM}$ denote the $L$-th order MACE embeddings of the optimized and target conformations, computed as described in Equation 3. For a given $L$, the Clebsch-Gordan coefficients $C^{LM}_{\eta_\nu,\mathbf{lm}}$ select the valid combinations of input angular momenta. While the second and third terms are not rotation-invariant, global alignment of the source configuration with the target ensures validity of the comparison.

This combined loss (weighted by $\alpha, \beta, \gamma$) drives the structural optimization by progressively aligning the descriptor representations of the initial and target altloc conformations. As illustrated in Fig. A.16, the backbone atom positions are recovered with high accuracy, but the side chains of glutamic acid (GLU 62) and lysine (LYS 63) adopt incorrect orientations. We hypothesize that this discrepancy arises because MACE features primarily capture the local chemical environment within a given radius, which cannot always resolve distinct side-chain rotamers.

These results suggest that while MACE features reliably capture backbone geometry, they are less sensitive to side-chain orientations. Recovering both backbone and side-chain configurations would require further exploration, but success in this direction could enable the use of MACE features (and other atomistic embeddings) for coarse-grained protein modeling with minimal loss of structural fidelity.

### J.2.2 Guiding AlphaFold3 reverse diffusion

We next investigate whether MACE descriptors can guide AlphaFold3's diffusion-based structure generation process. Specifically, we focus on proteins with alternate conformations (altlocs)—structural heterogeneity intrinsic to the protein within the unit cell. We analyze a subset of four proteins (Table B.15) exhibiting distinct backbone-separated altlocs, from Rosenberg et al. Rosenberg et al. (2024).

Starting from Gaussian noise, we guide AlphaFold3's reverse diffusion process to recover a target protein structure that minimizes a descriptor-based loss $\mathcal{L}$. The guided reverse SDE Song & Ermon (2019) is given by

$$d\mathbf{X} = -\left(\tfrac{1}{2}\mathbf{X} + \nabla_{\mathbf{X}} \log p_t(\mathbf{X} \mid \mathbf{a}) + \eta \nabla_{\mathbf{X}} \mathcal{L}\right)\beta_t dt + \sqrt{\beta_t}\mathbf{N},$$

where $\mathbf{X}$ are atomic coordinates, $\mathbf{a}$ is the amino acid sequence, $\beta_t$ is the noise schedule, and $\mathbf{N} \sim \mathcal{N}(\mathbf{0}, \mathbf{I})$. The term $\nabla_{\mathbf{X}} \log p_t(\mathbf{X} \mid \mathbf{a})$ is the unconditional score function, while $\nabla_{\mathbf{X}} \mathcal{L}$ provides descriptor-based guidance. The hyperparameter $\eta$ scales the guidance term.

To invert MACE embeddings, we define $\mathcal{L}$ as a discrepancy between the descriptors of the diffusion variable $\mathbf{X}$ and those of the target conformation $\mathbf{X}^{\text{target}}$ in the altloc region. The atomic environment is specified as all atoms within a 5 Å radius:

$$\mathcal{L} = - \begin{cases} \dfrac{1}{2} \left\| \mathbf{B}_0(\mathbf{X}_t) - \mathbf{B}_0^{\text{target}}(\mathbf{X}^{\text{target}}) \right\|_2^2, & \text{if } \left\| \mathbf{B}_0(\mathbf{X}_t) - \mathbf{B}_0^{\text{target}}(\mathbf{X}^{\text{target}}) \right\|_1 \leq \delta, \\ \delta \left( \left\| \mathbf{B}_0(\mathbf{X}_t) - \mathbf{B}_0^{\text{target}}(\mathbf{X}^{\text{target}}) \right\|_1 - \tfrac{1}{2}\delta \right), & \text{otherwise.} \end{cases} \tag{14}$$

Here, $\mathbf{B}_0$ is the zeroth-order invariant descriptor derived from $\mathbf{X}_t$, and $\mathbf{B}_0^{\text{target}}$ is the corresponding descriptor of the target conformation. The threshold $\delta$ switches the loss from L2 (small errors) to L1 (large errors). For all structures, we set $\eta = 0.5$ and $\delta = 0.25$.

As shown in Fig. A.14 A, unguided AlphaFold3 typically captures only one of the two alternate conformers. By guiding the reverse diffusion process with MACE descriptors, we can recover the alternate conformation that AlphaFold3 misses or preserve its original prediction. To quantitatively assess the distribution, we compute the normalized distance of each generated structure $\mathbf{X}$ relative to conformations $\mathbf{X}^{\text{A}}$ and $\mathbf{X}^{\text{B}}$ modeled in the PDB (Burley et al., 2017):

$$d_{\text{A}} = \left\| \mathbf{X} - \mathbf{X}^{\text{A}} \right\|_2^2$$
$$d_{\text{B}} = \left\| \mathbf{X} - \mathbf{X}^{\text{B}} \right\|_2^2$$
$$\text{Normalized Distance} = \left[ 1 - \min\left( \frac{d_{\text{B}}}{d_{\text{A}}}, \frac{d_{\text{A}}}{d_{\text{B}}} \right) \right] \cdot \text{sign}(d_{\text{A}} - d_{\text{B}}). \tag{15}$$

Negative values indicate proximity to conformation A, while positive values indicate proximity to conformation B. The resulting distributions (Fig. A.14 B) again show that guiding the diffusion process with descriptors shifts the distribution toward the mode AlphaFold3 misses or preserves the original distribution. Both results indicate that MACE descriptor-guided diffusion reliably steers AlphaFold3 toward accurate backbone geometries of both conformations in most cases.

## K Use of Large Language Models (LLMs)

The use of LLMs was restricted to minor proof-reading, stylistic polishing, and correcting typos in the text. All intellectual contributions, experiments, data analyses, and conclusions were executed and verified exclusively by the authors. Responsibility for the scientific content rests with the authors, and no part of the research process was delegated to LLMs.

