# OpenReview forum: "Representing local protein environments with machine learning force fields"
_ICLR.cc/2026/Conference — ICLR 2026 Poster_

### Official Review · Reviewer_fbDS · 2025-10-27

**Soundness:** 3
**Presentation:** 3
**Contribution:** 3
**Rating:** 6
**Confidence:** 2

**Summary:**

This paper explores the use of MLFFs as general-purpose representation learners for local protein environments. Instead of relying on sequence-based or handcrafted descriptors, the authors repurpose latent embeddings from pretrained MLFFs (AIMNet, MACE, OrbNet, Egret) as compact, physics-grounded descriptors for atomic neighborhoods in proteins. The study benchmarks MLFF embeddings on diverse downstream tasks—secondary structure and amino acid classification, pKa prediction, and NMR chemical shift regression—and shows that these representations outperform or rival specialized baselines such as PropKa and pKa-ANI. Overall, the paper establishes MLFFs as reusable, physics-informed foundation models for structural biology.

**Strengths:**

1. The central insight, repurposing MLFF embeddings as general-purpose, transferable protein environment representations, is both novel and timely. Most prior MLFF applications focus on energy or force prediction for small molecules; extending them to protein representation learning is original and valuable. The paper effectively bridges quantum-chemistry-based potentials and protein machine learning.

2. The experimental setup is comprehensive. The authors curate 165 k environments from 1048 proteins and evaluate four MLFF families on four biologically relevant tasks. Comparisons include classical and ML baselines (PropKa, pKa-ANI, UCBShift2-X). Results demonstrate meaningful improvements in both accuracy and interpretability. The inclusion of uncertainty quantification and physical consistency tests (e.g., ring-current effects) adds rigor.

3. The methodology that extracting embeddings from pretrained MLFFs and mapping them to canonical residue-centered environments is sound and clearly motivated. Statistical reporting (mean absolute errors, standard deviations) is adequate, and all experiments appear well-controlled.

4. The paper is well written and pedagogically organized. The introduction clearly motivates the challenge of representing local protein environments; figures (e.g., Fig. 1 and 2) effectively illustrate how embeddings are constructed and used. Terminology (canonical environment, focus residue, MLFF feature extraction) is consistent and accessible even to readers outside computational chemistry.

5. This work could substantially influence both computational biology and machine-learning communities by providing a physics-consistent alternative to sequence-only protein language models. MLFF embeddings encode quantum-derived information unavailable in existing representations and show transferability to tasks requiring local chemical precision. The idea of using pretrained MLFFs as “foundation models for atoms” could be broadly significant.

**Weaknesses:**

1. The study restricts environments to 5 Å radius regions; while suitable for local chemistry, it omits long-range electrostatic or conformational effects. For tasks like folding or binding prediction, this locality may be insufficient. A discussion or experiment extending to multi-scale contexts would strengthen generality claims.

2. MLFFs such as MACE and OrbNet require quantum-level pretraining on millions of molecules. Although embeddings are reused, the computational barrier to obtaining them limits accessibility compared to pretrained sequence models (e.g., ESM, ProtT5). The paper could better address scalability and efficiency trade-offs.

3. Qualitative case studies (e.g., helix → strand unfolding) are insightful but anecdotal. Quantitative metrics (e.g., correlation between embedding distances and RMSD/chemical similarity) would solidify interpretability claims.

4. While the authors compare to physics-based predictors, they do not benchmark against modern structure-aware geometric encoders (e.g., GVP-GNN, ProteinNeRF, FrameFold). Including such baselines would clarify whether MLFF embeddings provide advantages beyond standard geometric message-passing.

**Questions:**

1. Could the same MLFF-based representation transfer to nucleic acids or protein–ligand complexes, where local environments include non-canonical atoms and charges?

2. MLFFs have multiple internal layers encoding different orders of interaction (0th, 1st, 2nd). Did the authors investigate which layer yields the most informative embeddings for downstream tasks?

3. Since MLFF embeddings originate from networks with different scales and symmetries, how are they aligned or normalized across model families?

4. The authors mention uncertainty-aware predictions. Is uncertainty derived from ensemble variance, likelihood width, or another calibration technique?

5. Could MLFF embeddings serve as complementary features to pretrained protein language models, bridging physics-based and sequence-based representations?

---

> ### Author Response · Authors · 2025-11-20
>
> We thank the reviewer for the careful assessment and for the many constructive suggestions that point to promising extensions of this work. Several of the questions raised highlight opportunities we are excited to explore further.
>
> ## Environment size
> The behavior of the model with respect to the environment radius is an important point. MACE was trained on graphs constructed by taking, for each focus atom, all neighbors within a 6 Å radius. To stay consistent with this training distribution, the model internally truncates any provided environment to the same 6 Å neighborhood around each focus atom. In our implementation, we use a 5 Å cutoff and then include the entire residues for any atoms that fall within this distance, ensuring that every contributing residue is chemically complete.
>
> Following the reviewer's suggestion, we ablated the cutoff radius over values of 4, 5, and 6 Å (using the same residue-completion procedure in each case). A 5 Å cutoff yields the best performance across tasks. we added these results in the revised manuscript in Table B.8.
>
> This behavior is consistent with how environments are constructed. When the cutoff is increased, the resulting environment contains atoms that are no longer bonded to their natural partners. This introduces artificial degrees of freedom and changes the local energy landscape, producing chemically degraded contexts with unphysical free interactions. In contrast, using a smaller cutoff avoids these fragmented residues but reduces the contribution of longer range effects, limiting how much distant structural influence is reflected in the embedding.
>
> ## Computational efficiency
>
> We share the reviewer’s concern about computational cost. Our approach remains practical because it leverages pretrained MLFF models and only requires forward passes at inference time; we never retrain the MLFFs themselves. In this respect, computing MLFF embeddings is analogous to extracting intermediate activations from pretrained sequence models. In practice, descriptor computation is fast, parallelizable, and not a bottleneck in our pipeline.
>
> ## Qualitative case studies (e.g., helix → strand unfolding)
>
> Following the reviewer’s suggestion, we computed pairwise distances between structure embeddings and compared them to the pairwise RMSD matrix along the MD trajectory. To construct a structure embedding, we concatenated the Cα-centered local environment embeddings across all residues in the protein. As shown in Fig A.10, the resulting pairwise embedding distances are strongly correlated with the corresponding pairwise RMSD values, with Pearson and Spearman correlation coefficients of 0.92 and 0.98, respectively.
>
> ## Additional baselines
> We appreciate the suggestion to relate our method to modern structure-aware geometric encoders such as GVP-GNN, ProteinNeRF, and FrameFold. These models, however, are designed around different inputs and objectives than the setting we study here.
>
> GVP-GNN and related architectures operate primarily on the protein backbone and local backbone frames. In their standard formulation, they do not take explicit side-chain or hydrogen coordinates as input, reflecting their typical use in backbone-only design scenarios. ProteinNeRF and FrameFold similarly encode structure via frames derived from backbone atoms and focus on reconstructing or optimizing conformations in this reduced representation.
>
> As a consequence, these encoders omit the majority of atoms in a typical local environment. In our dataset, around 150–200 atoms appear within a given environment, of which only about 10 percent belong to the backbone. The remaining 90 percent—side chains and hydrogens—are essential for the fine-grained chemistry needed for pKa and chemical-shift prediction. Backbone or frame-only encoders treat environments with identical backbones as equivalent and thus cannot detect important variations such as aromatic-ring rotations (Fig. 4), which substantially affect chemical shifts while leaving the backbone unchanged.
>
> To address the reviewer’s request for additional learned baselines, we trained a message-passing neural network with learnable per atom type embeddings, optimized end to end on the same pKa and chemical shift tasks (Tables 1 and B5, also summarized in the general comment). This purely learned representation consistently underperformed the same architecture using frozen MLFF embeddings, supporting the benefit of the physics informed MLFF latent space. In addition, we compared against strong sequence and structure based protein language models from the ESM family, ESM 3, ISM ("Distilling Structural Representations into Protein Sequence Models."), and ESMFold. For each downstream task the best performing model was the MLFF based approach (Tables 1, B3, B4, and B5, also included in the general comment), further indicating that MLFF embeddings capture complementary, highly informative local chemistry.

---

> > ### Author Response · Authors · 2025-11-20
> >
> > ## Non-canonical atoms
> > MACE models can represent a broad range of chemical elements beyond C, H, N, and O, including atoms such as F, P, S, Cl, Br, and I. Thus, the representation is not intrinsically restricted to amino-acid chemistry and could, in principle, be used for nucleic acids or protein–ligand environments.
> >
> > While we did not evaluate these systems in the present work, extending MLFF based descriptors to nucleic acids and protein–ligand complexes, where non-canonical atoms and charges are prevalent, is a promising direction we intend to explore.
> >
> > ## Ablation study of different layers in MLFFs
> > In the MACE architecture, the descriptors we use already aggregate intermediate representations from both internal layers, so contributions from different interaction orders are present in the final embedding. For the other MLFF architectures, we originally used only the final-layer embeddings to maintain a consistent choice across models with many layers, and to avoid the impractical dimensionality that would result from concatenating all hidden states.
> >
> > Following the reviewer’s suggestion, we conducted an ablation comparing representations from different internal layers for both MACE and ORB on the chemical-shift prediction task (see Table B.9 and the general response). For MACE, concatenating the two hidden layers yields the best performance for most atom types. For ORB, the 10th hidden layer performs best across most cases, although we use the 15th layer in the main paper to maintain a consistent descriptor choice for MLFFs with many layers.
> >
> > ## Aligning different MLFF families
> > In the current work, embeddings from different MLFFs are used independently and are not aligned or normalized across models. Each downstream experiment uses a single MLFF variant. Investigating whether cross-model alignment or combined use of multiple force fields could yield additional benefits is an avenue we hope to examine in later work.
> >
> > ## Calculating likelihoods
> > Our approach to uncertainty quantification is based on a kernel-density–estimated likelihood computed in the MLFF embedding space. Environments whose embeddings fall in densely populated regions of the training data obtain high likelihoods and are treated as reliable, while embeddings in sparsely populated regions correspond to higher uncertainty. This method reflects how typical a given environment is relative to the training distribution, without relying on ensembles or calibration procedures.
> >
> > ## Incorporating MLFF embeddings with protein language models
> > The reviewer’s question about combining MLFF embeddings with protein language models highlights a particularly exciting direction. MLFF embeddings require an explicit 3D structure and therefore cannot be used at inference time for sequence-only models when structures are not available. However, when structures are available, during training or in curated datasets, MLFF features and PLM embeddings can be integrated, providing complementary perspectives: MLFFs supply chemically detailed local structure, while language models contribute broad evolutionary and sequence-level context. Exploring such combined representations could open up compelling opportunities for downstream applications.

---

### Official Review · Reviewer_RxQj · 2025-10-31

**Soundness:** 3
**Presentation:** 3
**Contribution:** 2
**Rating:** 4
**Confidence:** 4

**Summary:**

Local protein environments contain highly relevant chemistry that affects its function and interactions. This paper benchmarks existing machine learning force field (MLFF) methods on their ability to understand local protein structures. They evaluate a suite of MLFFs on their ability to predict the protonation state, secondary structure, and amino acid types.

**Strengths:**

This paper rigorously evaluates multiple popular MLFF methods on across benchmarks and connect them to some protein structure related tasks.
They perform an interesting analysis on the chemical shift prediction.

**Weaknesses:**

The evaluation benchmarks such as secondary structure, amino acid type prediction are a bit straightforward.

While MLFF naturally learn the physics of local structural environments, there are other machine learning based approaches that reason over the local structure and are suitable to predict protonation state, secondary structure, and amino acid types. This work does not benchmark MLFF versus these methods on representing local structure.

[1] Simulating 500 million years of evolution with a language model. Hayes et al.
[2] 3D deep convolutional neural networks for amino acid environment similarity analysis. Torng et al.
[3] Distilling Structural Representations into Protein Sequence Models. Ouyang-Zhang et al.

**Questions:**

Are there any other interesting local protein structure properties to evaluate on?

How well do non-FF based local structure models, such as ESM3, perform on these benchmarks?

---

> ### Author Response · Authors · 2025-11-20
>
> We thank the reviewer for their careful reading of our work and for noting the value of our systematic benchmarking of multiple MLFFs on local protein-structure tasks and our analysis of chemical shift prediction, and we address the remaining concerns below.
>
> ## Secondary structure and amino acid type prediction benchmarks.
>
> We appreciate the reviewer’s point that some of our analyses are more illustrative than practically oriented, and we have revised the text to spell out this distinction. In the updated manuscript, the amino-acid identity and secondary-structure experiments in Section 4 are explicitly described as lightweight "probing tasks", not as intended end-use scenarios. Their purpose is to provide intuition: they show that embeddings learned purely from DFT energies and forces nevertheless encode information about residue type and local structural context.
>
> ## Other interesting local protein structure properties to evaluate on?
>
> Indeed, there are additional experimental observables that reflect the underlying chemistry, structure, and dynamics of local protein environments. Particularly attractive for immediate exploration are NMR-derived measurements such as order parameters ($S^2$) and relaxation rates ($R_1$, $R_2$), which capture local mobility and dynamics, as well as local thermodynamic stability parameters (local $\Delta G$) obtained from hydrogen–deuterium exchange (HDX-MS) experiments.  Beyond local structural properties, it is also interesting to explore MLFF-based representation to characterize, predict, and design local biomolecular interactions, e.g., protein-protein, protein-ligand, and protein-ion systems. We are currently working on these directions in an ongoing work.
>
>
> ## Additional baselines
>
> As detailed in our general response on the updated baselines, we have revised the manuscript and added methods [1] and [3] as non-MLFF baselines for all tasks: pKa, secondary structure prediction, residue type prediction, and chemical shift prediction- reported in Table 1, Table B.3, Table B.4, and Table B.5. For ESM-3, we evaluate both the sequence only and sequence+structure variants. Overall, while these non-FF representations perform reasonably well, the best performing model on every benchmark is MLFF-based.
>
>
> Regarding the specific suggestion to benchmark against Torng et al. [2], we now clarify in the paper that this 3D-CNN is architecturally quite different from the non-MLFF encoders we evaluate. It operates on coarse voxel grids of only heavy atoms (C/N/O/S) around a residue and is trained as a task-specific amino acid classifier, rather than providing reusable local environment embeddings; it also omits hydrogens, which we show are crucial for predicting chemical shifts. For these reasons, we treat it as a complementary line of work and do not include it as a direct baseline.

---

### Official Review · Reviewer_dp5R · 2025-11-01

**Soundness:** 1
**Presentation:** 3
**Contribution:** 2
**Rating:** 4
**Confidence:** 3

**Summary:**

This paper proposes repurposing intermediate embeddings from pre-trained Machine Learning Force Fields (MLFFs) as "physics-grounded" feature representations for local protein environments. The authors benchmark embeddings from MACE, AIMNet, and others, claiming state-of-the-art (SOTA) results in $pK_{a}$ prediction (reportedly outperforming pKa-ANI) and NMR chemical shift prediction (reportedly outperforming UCBShift2-X). The method's physical realism is validated via case studies, such as aromatic ring current effects.

**Strengths:**

1. Repurposing MLFFs as "foundation models" for structural biology is in nnovative and valuable.
2. The design of the validation experiments (e.g., ring current effect , helix unfolding ) is a commendable standard for physical realism.
3.The paper effectively demonstrates the general-purpose nature of the embeddings for zero-shot clustering and generative guidance.

**Weaknesses:**

1.The paper's central claims rest on comparisons against pKa-ANI and UCBShift2-X that appear to be factually incorrect or based on flawed implementations.

2.The $pK_{a}$ evaluation is missing the entire 2024/2025 SOTA, invalidating its performance claims.

3.The paper compares its structural embeddings against sequence (ESM) embeddings. It critically fails to benchmark against the most obvious and relevant competitors: other structural embeddings, namely those from the AlphaFold2 or ESMFold structure modules.

**Questions:**

see weakness

---

> ### Author Response · Authors · 2025-11-13
> **Question regarding baselines**
>
> We thank the reviewer for their review, we wanted to quickly follow-up regarding two points that the reviewer mentioned:
>
> 1. They mention that our "comparisons against pKa-ANI and UCBShift2-X to be factually incorrect or based on flawed implementations". Could the reviewer kindly elaborate why they believe this is the case?
> 2. With regard to $pK_a$ evaluation: could the reviewer provide citations for the baselines we missed?

---

> ### Author Response · Authors · 2025-11-20
>
> We thank the reviewer for their thoughtful comments and appreciate the positive notes about the physical realism of our case studies and the general-purpose nature of the embeddings.
>
> ## Comparison vs pKa-ANI
> We thank the reviewer for raising this important point and for prompting us to revisit how we compare our method to pKa-ANI. Upon re-examining our setup, we realized that a key source of mismatch is the training data: our models were trained on simulated pKa values computed over AFDB structures, whereas pKa-ANI was trained on PDB structures with experimental pKa measurements from the PKAD dataset (for which the authors did not report the exact train–test split). Since then, PKAD has been updated through PKAD-2 to the curated PKAD-R version.
>
> To enable a more fair comparison, we re-evaluated our models on the PDB structures corresponding to all entries in PKAD-R and compared our predictions to the experimental pKa values, and compared them with the publicly available pKa-ANI model. This protocol is strongly biased **in favor of pKa-ANI**, as it was trained in exactly this setting (PDB structures and experimental PKAD values). We noticed that even under these pKa-ANI–favorable conditions, our models achieve lower errors for three of the four residue types considered; for the remaining residue type, the weaker performance is plausibly due to the substantially smaller number of training examples available for that class in our training data. We have updated the manuscript to report these revised and more accurate comparisons, and, as noted in our response to the general comments, we have added a summary table of comparisons in Table B.7 that clearly reflects these updated results.
>
> ## Comparisons to UCBShift2
>
> As described in the Data pargraph in Section 3, the data for the chemical shift prediction experiments were sourced from RefDB, a database of carefully curated re-referenced experimental chemical shifts originally sourced from the biomolecular resonance database (BMRB). Training on chemical shifts from RefDB is common practice as employed in UCBShift2, and in fact, recommended as it factors out experiment/instrument-specific biases.
>
> **Evaluation protocol.** The train and test split is constructed by sequence-similarity--based clustering of the dataset, to minimize the overlap between train and test sets. For comparison, we ran the UCBShift2 by providing the AF2 structures on the test set. This evaluation is biased in favor of UCBShift as we did not exclude test data points that may have been part of UCBShift's training set.
>
> [Ptaszek, Aleksandra L., et al., 2024] Ptaszek, Aleksandra L., et al. "UCBShift 2.0: Bridging the gap from backbone to side chain protein chemical shift prediction for protein structures." Journal of the American Chemical Society 146.46 (2024): 31733-31745.

---

> > ### Author Response · Authors · 2025-11-20
> >
> > ## Justification for using AlphaFold structures
> >
> > We use AlphaFold structures to predict shifts for the following critical reasons.
> >
> > **Using experimental structures limits generalization.** Firstly, and most crucially, using PDB structures sourced from X-ray crystallography creates a fundamental limitation: our method would not generalize at inference time to protein sequences where experimental crystal structures are unknown. This severely restricts the practical applicability of the approach to the vast majority of protein sequences that lack experimental structural data.
> >
> > **Crystallographic structures represent different experimental conditions than NMR.** Beyond this primary concern, it is essential to recognize that crystallography and NMR represent fundamentally different experimental settings. It is not always the case that exactly the same sequence is characterized both in NMR spectroscopy and X-ray crystallography. This reality already omits the significant environmental differences between crystal-state and solution-state conditions, which can substantially affect protein structure and dynamics.
> >
> > **Lack of one-to-one correspondence between PDB and BMRB.** We computed the sequence similarity between the FASTA sequence associated with each BMRB ID and the FASTA sequence of the corresponding crystal structure, finding that approximately 15.8% of amino acids in the sequences were misaligned. Because we are interested in local chemical environments, even minor changes in sequences can result in significant alterations to chemical shift properties. Furthermore, a single sequence in RefDB is often associated with multiple PDB entries, creating inconsistency and ambiguity when selecting a reference structure for descriptors. For instance, BMRB ID 15064 corresponds to 16 different PDB entries, while BMRB ID 15726 is linked to 7 distinct PDB entries. When the same BMRB ID is associated with multiple PDB entries, it becomes unclear which structure should be chosen as the reference.
> >
> > AlphaFold helps alleviate these issues. To mitigate these issues and ensure a consistent and controlled source of structural information, we utilize AlphaFold (AF) predictions. We understand that AlphaFold might introduce certain biases, but these biases are at least systematic and consistent across all sequences, including previously unseen ones. This approach provides uniform structural predictions for any protein sequence, ensuring both consistency in our training data and broad applicability at inference time.

---

> > > ### Author Response · Authors · 2025-11-20
> > >
> > > ## Baselines
> > > We thank the reviewer for pointing out the importance of comparing against AlphaFold-2 and ESMFold-based representations.
> > >
> > > **AF2 IPA.** In the AlphaFold2 architecture, the IPA module is designed as a structure generator: it takes sequence derived features together with local backbone frames and iteratively refines them into global backbone frames and side chain dihedral angles, which are then used to build the final three dimensional structure. In other words, IPA consumes sequence based internal states and produces structure, rather than ingesting an existing structure to produce a reusable structural embedding. For this reason, IPA is more naturally viewed as a *structure decoder* than as a general purpose structure encoder.
> > >
> > > **AF2 Evoformer.** The most relevant AlphaFold2 representations for property prediction are therefore the Evoformer embeddings, which are also fundamentally sequence derived. While we did not explicitly evaluate Evoformer features, we did benchmark ESM2 embeddings, which have been shown to match or surpass Evoformer representations on a range of downstream tasks [Wayment-Steele et al., 2025]. Thus, our comparison to ESM2 provides a strong proxy for what one would expect from AlphaFold2 sequence based internal states.
> > >
> > > **Limitations of AF2-based embeddings.** However, because these internal representations originate from sequence alone, they are inherently limited for the types of tasks we study. In particular, they are not fully sensitive to fine grained side chain rearrangements, such as the rotated aromatic rings in phenylalanine highlighted in Figure 4, that can substantially alter local electrostatics and consequently pKa values and chemical shifts. In that respect, they share similar limitations with methods that only encode backbone geometry.
> > >
> > >
> > > **ESMFold embeddings.** Following reviewer's suggestion, we implemented the comparison to ESMFold embeddings, and incorporated the results into the revised manuscript, together with a new summary table of method comparisons discussed in our general response. Specifically, we extracted residue-level embeddings from the final layer of the ESMFold folding trunk and used them as inputs to the same downstream predictors and with the same structural inputs as in our MLFF-based evaluations. Across all property prediction tasks, the best performing model was always one of the MLFF-based representations, with ESMFold consistently performing worse. We believe this analysis directly addresses the reviewer’s concern and further highlights the advantage of MLFF-derived embeddings for fine-grained local property prediction.
> > >
> > > **ESM-3 and ISM.** In addition, we have added comparisons to structure-conditioned protein sequence models, ESM-3 [Hayes, Thomas, et al., 2025] and ISM [Ouyang-Zhang, Jeffrey, et al., 2024], which are more relevant to the questions posed by the reviewer. We report the accuracy of these models for all tasks: pKa, secondary structure prediction, residue type prediction, and chemical shift prediction—reported in Table 1, Table B.3, Table B.4, and Table B.5, respectively. For ESM-3, we evaluate both the sequence only and sequence+structure variants. Overall, MLFF-based models consistently surpass local structural–descriptor baselines in most settings, and for all tasks considered the best-performing model is MLFF-based.
> > >
> > >
> > > Our interpretation is that sequence derived embeddings excel at capturing global and evolutionary context, but they lack the precise local structural information required for high resolution tasks such as pKa and chemical shift prediction. This is exactly the gap our structural embeddings are designed to fill, and we believe this clarifies why we did not benchmark against the AlphaFold2 structure module directly.
> > >
> > > We refer to Table B.5 in the manuscript and to the general response for a summary of these updated comparisons.
> > >
> > > [Hayes, Thomas, et al., 2025] Hayes, Thomas, et al. "Simulating 500 million years of evolution with a language model." Science 387.6736 (2025): 850-858.
> > >
> > > [Ouyang-Zhang, Jeffrey, et al., 2024] Ouyang-Zhang, Jeffrey, et al. "Distilling structural representations into protein sequence models." bioRxiv (2024): 2024-11.
> > >
> > > [Wayment-Steele et al., 2025] Wayment-Steele et al., *Learning millisecond protein dynamics from what is missing in NMR spectra*, bioRxiv, March 2025. [https://www.biorxiv.org/content/10.1101/2025.03.19.642801v1.abstract](https://www.biorxiv.org/content/10.1101/2025.03.19.642801v1.abstract)

---

### Official Review · Reviewer_AnCg · 2025-11-01

**Soundness:** 3
**Presentation:** 3
**Contribution:** 3
**Rating:** 6
**Confidence:** 4

**Summary:**

This work proposes to leverage pretrained ML interatomic potential embeddings as canonical descriptors of local protein environments. Embeddings are extracted from a residue-centered neighborhood. Various applications including pKa prediction of titrable residues and protein NMR chemical shift prediction are explored, with MLFF-based embeddings show good representative power for downstream tasks.

**Strengths:**

- Repurposing latent features of MLFFs as canonical protein descriptors is a timely, well-motivated idea that links quantum-level atomistic modeling with biomolecular representation learning.
- The paper covers a solid range of downstream tasks tied to experimental observables and provides thorough analysis of the physical plausibility of its predictions.

**Weaknesses:**

- Dataset and baselines: For the pKa and NMR shift tasks, the baselines are evaluated in conditions that differ from their intended use, whereas the MLFF-feature models introduced here are trained directly for the target objective. For instance, the pKa baselines are designed to predict experimental values, while the proposed methods are trained to reproduce a cheaper computational reference. This creates a benchmark mismatch, since the baselines are not optimized for the reference chosen in this work. Because the core question is whether MLFF features are useful, a more appropriate primary comparison would be a standard GCN with simple learned embeddings rather than prior task-specific baselines. This issue is compounded by the use of AFDB structures as inputs.
- Experimental design: Some experiments are not very indicative of how the embeddings would be used in practice. Inferring amino acid type or secondary structure from full atomic coordinates (Section 4) is a trivial task under those inputs, and the distribution-shift analysis in Section 5 (Fig. 2) mainly shows that embeddings from energy-relaxed structures are similar, which is expected since both MLFFs and classical force fields approximate the same physical energy landscape. Several experiments currently placed in the appendix appear more compelling and better aligned with the paper’s motivation. I recommend reorganizing the manuscript so that the most informative use cases are in the main text, especially given that all experiments are listed as contributions but not all are presented in the main body.

**Questions:**

- How sensitive are results to the MLFF layer chosen for embeddings?
- MACE is an architecture, but there are several variants according to the dataset it is trained on. It seems like the MACE-OFF23 is used in this work, could authors elaborate?
- For AF3-guided structure selection in Fig A14, could authors report the optimization statistics similarly to Fig A13? (What fraction of optimization result in the target structure, or whether it always lead to the target structure)

---

> ### Author Response · Authors · 2025-11-20
>
> We thank the reviewer for the careful assessment. We are glad that the reviewer appreciated the use of MLFF-derived embeddings for representing local protein environments, as well as the depth and physical grounding of our evaluation.
>
> ## Dataset and baselines
>
> ### Simple GCN baseline.
> As suggested by the reviewer, we trained our GCN model using standard learned embeddings (optimized end-to-end) to predict pKa and chemical shift (backbone atoms) values until convergence. In this baseline, the input for each local atomic environment (described in Section 3) consists of the atom types and their positions. The coordinates are first mapped to a sinusoidal positional encoding, which is then concatenated with the embedded atom-type identifiers to produce the node features for the GCN. This learned-embedding baseline performs worse than the MLFF-embedding model, highlighting the utility of the proposed features. The comparison is presented in Tables 1 and B5 in the manuscript and in the general response.
>
> ### Dataset: pKa prediction
>
> We thank the reviewer for correctly pointing out the benchmark mismatch; this is indeed an important issue that was, indeed, overlooked by us.
>
> In the revision, we have addressed this issue in two ways:
> 1. _Clarity in text._ We clarified in the caption of the pK$_\text{a}$ results table (Table 1) that the comparison is based on simulated values, for which PropKa and pKa-ANI were not optimized to fairly depict the results.
> 2. _Added comparisons._ To fairly assess our models, we compared our models on the experimental datasets used by PropKa and pKa-ANI models. Because these experimental datasets are too small to train on, we evaluate using our models trained on simulated values (and AFDB structures), and report their performance on the corresponding PDB structures:
>
>
>     - *PropKa comparison:* PropKa was trained on experimental pKas sourced from 85 pKa entries. In their work, the authors provided experimental values and predictions only for these 85 training entries and mention an additional 201-entry test set, composed of datasets from three prior studies, without releasing its experimental values, model predictions, or PDB identifiers. As a practical compromise, we evaluated our pKa model (trained on simulated values and AFDB structures) on the PDB structures corresponding to the 85 PropKa _training entries_. This setup is inherently biased in favor of PropKa, yet our model attains lower error on more than 50% of the residue–protein pairs. The results of this comparison are summarized in Table~B.6 in the manuscript and in the general response.
>
>      - *pKa-ANI comparison:* pKa-ANI was originally trained and evaluated on the PKAD dataset, but the authors do not report their exact train–test split. Since the publication of pKa-ANI, PKAD has been updated to PKAD-2 and, more recently, to the curated PKAD-R version. For our evaluation, we applied both the published pKa-ANI model and our own models (trained on simulated values and AFDB structures) to the PDB structures corresponding to all entries in PKAD-R. Although this protocol is biased in favor of pKa-ANI, which was trained on closely related data, our model attains lower error for 3 out of the 4 residue types considered. For the remaining residue type, our weaker performance is plausibly explained by the substantially smaller number of training examples available during training. The results of this comparison are summarized in Table~B.7 in the manuscript and in the general response.
>
>
> ### Dataset: Chemical shift prediction
>
> As described in the Data paragraph in Section 3, the data for the chemical shift prediction experiments were sourced from RefDB, a database of carefully curated re-referenced **experimental chemical shifts** originally sourced from the biomolecular resonance database (BMRB). Training on re-referenced chemical shifts is common practice [Ptaszek, Aleksandra L., et al., 2024], and in fact, recommended as it factors out experiment/instrument-specific biases.
>
> [Ptaszek, Aleksandra L., et al., 2024] Ptaszek, Aleksandra L., et al. "UCBShift 2.0: Bridging the gap from backbone to side chain protein chemical shift prediction for protein structures." Journal of the American Chemical Society 146.46 (2024): 31733-31745.

---

> > ### Author Response · Authors · 2025-11-20
> >
> > ### The use of AlphaFold structures
> >
> > Several critical issues arise when attempting to use experimental protein structures from the Protein Data Bank (PDB) for our analysis.
> >
> > **Using experimental structures limits generalization.** Firstly, and most crucially, using PDB structures sourced from X-ray crystallography creates a fundamental limitation: our method would not generalize at inference time to protein sequences where experimental crystal structures are unknown. This severely restricts the practical applicability of the approach to the vast majority of protein sequences that lack experimental structural data.
> >
> > **Crystallographic structures represent different experimental conditions than NMR.** Beyond this primary concern, it is essential to recognize that crystallography and NMR represent fundamentally different experimental settings. It is not always the case that exactly the same sequence is characterized both in NMR spectroscopy and X-ray crystallography. This reality already omits the significant environmental differences between crystal-state and solution-state conditions, which can substantially affect protein structure and dynamics.
> >
> > **Lack of one-to-one correspondence between PDB and BMRB.** We computed the sequence similarity between the FASTA sequence associated with each BMRB ID and the FASTA sequence of the corresponding crystal structure, finding that approximately 15.8% of amino acids in the sequences were misaligned. Because we are interested in local chemical environments, even minor changes in sequences can result in significant alterations to chemical shift properties.
> >
> > Furthermore, a single sequence in RefDB is often associated with multiple PDB entries, creating inconsistency and ambiguity when selecting a reference structure for descriptors. For instance, BMRB ID 15064 corresponds to 16 different PDB entries, while BMRB ID 15726 is linked to 7 distinct PDB entries. When the same BMRB ID is associated with multiple PDB entries, it becomes unclear which structure should be chosen as the reference.
> >
> > AlphaFold helps alleviate these issues. To mitigate these issues and ensure a consistent and controlled source of structural information, we utilize AlphaFold (AF) predictions. We understand that AlphaFold might introduce certain biases, but these biases are at least systematic and consistent across all sequences, including previously unseen ones. This approach provides uniform structural predictions for any protein sequence, ensuring both consistency in our training data and broad applicability at inference time.
> >
> > ## Experimental design
> > We thank the reviewer for pointing this out. We agree that not all experiments are equally indicative of how the embeddings would be used in practice. We revised the manuscript to make this distinction explicit. The amino-acid identity and secondary-structure experiments in Section 4 are now clearly framed as simple "probe tasks" rather than realistic downstream applications; they serve as motivation by showing that embeddings trained only on DFT energies and forces already encode residue identity and local structure. We also clarify that the distribution-shift analysis in Section 5 is intended as a sanity check of sensitivity to energy-relaxed versus experimental structures, where some similarity is expected for embeddings approximating the same energy landscape. Finally, we have reorganized the paper so that the previous Fig. 2 has been moved to the appendix and the UMAP visualization (formerly Fig. A.9) has been moved into the main text, in line with the reviewer’s suggestion. If the reviewer has any further suggestions on specific experiments to be more informative as a reader, we would be glad to move them the main text and reorganize accordingly.

---

> ### Author Response · Authors · 2025-11-20
>
> ## MLFF embeddings from different layers
> Following the reviewer's suggestion, we conducted an ablation study of chemical shift prediction using different MACE and ORB layers, it is added to the updated manuscript (see Table B.9).
> MACE contains two message-passing layers: the descriptors used in the main paper are the concatenation of the invariant features from both layers (following the recommendation in MACE-OFF23; default setting on the MACE codebase). In the ablation, we further evaluated performance using the invariant features from each layer separately.
> ORB contains 15 message-passing layers. In the paper, we used the embeddings from the final hidden layer to maintain a consistent descriptor configuration across MLFFs with many layers, as concatenating all hidden states would produce impractically high-dimensional features. In the ablation, we additionally report chemical-shift prediction performance using embeddings from the 5th, 10th, and 15th layers.
> The results of this ablation appear in Table B.9. For MACE, concatenating the two hidden layers yields the strongest results for most atom types, while for ORB the 10th layer performs best across most cases, even though the 15th layer is used in the main paper for consistency. Notably, even when using the best-performing ORB descriptors from the 10th layer, all MACE variants still outperform all ORB variants across atom types.
>
> ## Choice of MACE model
> In our work we used the MACE-OFF23 model because, among the available models, it is the only one trained on organic molecules and dipeptides, and thus has the training set that closely matches the biomolecular chemistry.
>
> ## Inverting MACE descriptors via optimization
> The goal of this experiment is to start from the experimentally refined coordinates of one alternate conformation (altloc) and adjust them so that their MLFF embedding matches that of a second altloc for the same residues. Concretely, we initialized the model with the coordinates of one altloc and optimize these coordinates to align their MLFF embedding with that of the other altloc. Because this is a fully deterministic optimization procedure rather than a stochastic guidance process, it always converges to the same solution given the same initialization. For completeness and direct comparability to our (stochastic) guidance setting, we now report optimization results on the same set of proteins used in the guidance experiments; these results are shown in Figure A.15.

---

### Author Response · Authors · 2025-11-20
**General Response Part 1**

We thank all reviewers for their thoughtful and constructive feedback. Several common themes emerged across the reviews, in particular, the relationship between simulated and experimental targets, baseline coverage and ablation studies. We summarize our main clarifications and new experiments on these shared points below, and we have updated the manuscript accordingly so that the changes are visible there as well.

## Comparisons with experimental values

**Critique.** Reviewers AnCg and dp5R raised concerns about mismatches between our simulated training targets and experimental baselines.

**Response.**
For pKa, we clarified in the text that Table 1 reports results on **simulated** targets, which makes the comparison conservative and biased against methods trained on experimental values, and we added new evaluations on the experimental datasets associated with PropKa and pKa-ANI (Tables B.6 and B.7). For PropKa, we compare on the experimental set used for its training; for pKa-ANI, we evaluate on the updated PKAD-R dataset, noting that the original work did not publish its exact train–test split. In both cases, the protocols are biased in favor of the baselines, yet our model outperforms PropKa on more than 50% of its entries and pKa-ANI on 3 of the 4 residue types considered. The comparison results are presented in the tables below.
For chemical shifts, our model is trained directly on **experimental** values from RefDB with sequence-similarity–based train/test clustering, and our comparison to UCBShift2 is biased in UCBShift2’s favor because we did not remove RefDB entries that may have been included in its training set.

| PDB ID | Glu acid PropKa | Glu acid (Ours) | Asp acid PropKa | Asp acid (Ours) |
|--------|------------------|----------------------|-----------------|----------------------|
| 1PGA:A | 0.413            | **0.294**            | 0.362           | **0.292**            |
| 1IGD:A | 0.489            | **0.182**            | 0.352           | **0.274**            |
| 1A2P:A | **0.525**        | 1.046                | 0.426           | **0.403**            |
| 4ICB:A | 0.579            | **0.546**            | **0.299**       | 1.018                |
| 1BEO:A | –                | –                    | 0.730           | **0.511**            |
| 4LZT:A | **0.629**        | 0.665                | 0.488           | **0.404**            |
| 3RN3:A | 0.548            | **0.462**            | 0.423           | **0.412**            |
| 2RN2:A | 0.274            | **0.185**            | 0.683           | **0.585**            |
| 2OVO:A | 0.377            | **0.376**            | **0.180**       | 1.571                |
| 1XNB:A | **1.001**        | 1.270                | **0.436**       | 1.020                |
| 135L:A | 0.470            | 1.700                | **0.480**       | 1.043                |


| Amino acid     | pKa-ANI | Ours  |
|----------------|---------|-------------|
| Glutamic Acid  | 0.696   | **0.695**   |
| Aspartic Acid  | 0.978   | **0.831**   |
| Lysine         | 0.648   | **0.360**   |
| Histidine      | **0.899** | 0.925     |

---

> ### Author Response · Authors · 2025-11-20
> **General Response Part 2**
>
> ## Baseline Coverage
>
> **Critique.** Reviewers AnCg, dp5R, RxQj, and fbDS asked for additional baselines.
>
> **Response.**
> - We added comparisons to strong protein language–model baselines, including ESM-3, ISM, and ESMFold structural embeddings. Across all downstream tasks (Tables 1, B.3, B.4, and B.5), the best-performing model is always an MLFF-based representation.
> - To isolate the benefit of MLFF embeddings, we trained an end-to-end GCN with purely learned embeddings. As shown in Tables 1 and B.5, this model consistently underperforms its MLFF-initialized counterpart, highlighting the utility of the MLFF features.
> - For chemical shifts, these additional baselines are currently reported for backbone atoms only, due to computational and time constraints in recomputing all models on the full set of atom types; we will extend this table to side-chain atoms.
>
>
> Key results for these additional baselines are reported in the tables below.
>
> **pKa:**
>
> ESM-Features + GCN: ESM3 (Seq), ISM, ESM (Struc), ESMFold
>
> MLFF-Features + GCN (Ours): MACE, OrbNet, AIMNet, Egret
>
> | Learned | ESM3 (Seq) | ISM   | ESM (Struc) | ESMFold | MACE  | OrbNet | AIMNet    | Egret |
> | ------- | ---------- | ----- | ----------- | ------- | ----- | ------ | --------- | ----- |
> | 0.518   | 0.459      | 0.382 | 0.360       | 0.351   | 0.306 | 0.306  | **0.265** | 0.304 |
> | 0.582   | 0.528      | 0.430 | 0.388       | 0.419   | 0.280 | 0.284  | **0.267** | 0.272 |
> | 0.652   | 0.359      | 0.304 | 0.295       | 0.278   | 0.320 | 0.282  | **0.270** | 0.298 |
> | 0.615   | 0.488      | 0.428 | 0.408       | 0.383   | 0.424 | 0.441  | **0.380** | 0.440 |
>
> **Chemical shifts**:
>
> ESM-Features + GCN: ESM3 (Seq), ISM, ESM3 (Struc), ESMFold
>
> MLFF-Features + GCN (Ours): MACE, Egret, Orb, AIMNet
>
> | Atom | Learned | ESM3 (Seq) | ISM   | ESM3 (Struc) | ESMFold | MACE      | Egret     | Orb   | AIMNet |
> | ---- | ------- | ---------- | ----- | ------------ | ------- | --------- | --------- | ----- | ------ |
> | HA   | 0.405   | 0.235      | 0.220 | 0.193        | 0.203   | 0.180     | **0.176** | 0.200 | 0.201  |
> | H    | 0.528   | 0.393      | 0.377 | 0.600        | 0.365   | 0.300     | **0.295** | 0.333 | 0.325  |
> | CA   | 1.753   | 1.049      | 0.814 | 0.667        | 0.667   | **0.584** | 0.599     | 0.683 | 0.660  |
> | N    | 3.174   | 2.349      | 2.275 | 1.845        | 1.986   | **1.642** | 1.667     | 2.015 | 1.875  |
> | CB   | 1.326   | 1.092      | 0.884 | 0.758        | 0.775   | 0.716     | **0.705** | 0.792 | 0.780  |
> | C    | 1.517   | 1.038      | 0.923 | 0.842        | 0.876   | 0.744     | **0.743** | 0.847 | 0.824  |
>
>
>
> ## Ablation studies
>
> **Critique.** Reviewers AnCg and fbDS asked for ablation studies on (i) the radius of the local protein environment and (ii) the choice of internal MLFF layer.
>
> **Response.**
> - For backbone chemical shifts, an ablation on the environment radius (4/5/6 Å), reported in Table B.8, shows that a 5 Å cutoff performs best. Larger cutoffs introduce chemically inconsistent partial residues, while smaller cutoffs omit relevant context.
> - A layer-wise ablation for MACE and ORB on backbone chemical-shift prediction (Table B.9) shows that MACE performs best with its default concatenated layers, whereas ORB performs best around layer 10. Notably, *all* MACE variants outperform *all* ORB variants across atom types. We nevertheless retain final-layer descriptors for ORB in the main text to maintain a consistent descriptor choice across MLFFs. The per-atom MAEs for this ablation are summarized in the table below.
>
>
> | Atom | MACE L1 | MACE L2 | MACE L1+2 | Orb L5 | Orb L10 | Orb L15 |
> |------|---------|---------|-----------|--------|---------|---------|
> | C    | 0.797   | 0.755   | **0.744** | **0.836** | **0.836** | 0.847   |
> | CA   | 0.619   | 0.593   | **0.584** | 0.678  | **0.660** | 0.683   |
> | CB   | 0.756   | **0.705** | 0.716   | 0.778  | **0.777** | 0.792   |
> | H    | 0.318   | **0.300** | **0.300** | **0.330** | 0.331  | 0.333   |
> | HA   | 0.191   | **0.178** | 0.180   | 0.198  | **0.196** | 0.200   |
> | N    | 1.771   | 1.666   | **1.642** | 1.929  | **1.889** | 2.015   |

---

### Meta-Review · Area_Chair_nYLr · 2026-01-07

**Summary:**

This paper proposes using intermediate embeddings from pretrained machine learning force fields (MLFFs) as representations of local protein environments and demonstrates their effectiveness across several downstream tasks. Reviewers generally agreed that the idea is timely, well motivated, and technically sound, and that the empirical evaluation is extensive and carefully carried out.

While several reviewers raised concerns about baseline comparisons, experimental design choices, and clarity of certain tasks, these concerns were directly addressed in the rebuttal through additional experiments, new baselines, expanded ablation studies, and clearer framing of the results. After reading the paper and carefully checking the rebuttal, I find that the authors have adequately addressed the substantive issues raised by reviewers, and the work meets the bar for acceptance.

**Reviewer Concerns:**

The main concerns raised by reviewers focused on three areas: (1) comparisons to experimental versus simulated baselines, (2) missing sequence- and structure-based representation baselines, and (3) the practical relevance of some evaluation tasks. The authors provided detailed responses clarifying the evaluation protocols, added new comparisons against experimental datasets, and incorporated additional strong baselines, including ESM-based and ESMFold representations. These strengthen the empirical claims and the technical contribution compared to prior work.

Reviewers also questioned whether some tasks (e.g., amino-acid identity or secondary structure prediction) were too simple to demonstrate real-world utility. The rebuttal appropriately reframed these as probing tasks rather than primary applications and reorganized the manuscript to emphasize the more compelling downstream use cases. Additional ablations and analyses further improved clarity around representation choices and robustness. Overall, after reviewing the rebuttal, I believe the remaining concerns are either resolved or appropriately acknowledged as limitations.

**Reviewer Scores:**

The reviewer scores were 6 (AnCg), 6 (fbDS), 4 (dp5R), and 4 (RxQj).

Based on the discussion and the detailed rebuttal, the reviewers with higher initial scores (AnCg and fbDS) would very likely have maintained their positive assessments, as their main concerns were addressed through additional baselines, clearer evaluation protocols, and expanded ablation studies. For the reviewers with borderline scores (dp5R and RxQj), even if they had the opportunity to participate fully in the discussion and carefully read the rebuttal, their scores would likely have increased slightly. The rebuttal directly addressed their concerns regarding baseline coverage, experimental framing, and comparison to structure- and sequence-based representations.

---

### Decision · Program_Chairs · 2026-01-26

Accept (Poster)